# Disentangling Misreporting from Genuine Adaptation in Strategic Settings: A Causal Approach

**Dylan Zapzalka**[1]  **Trenton Chang**[1]  **Lindsay Warrenburg**[2]  **Sae-Hwan Park**[2]
**Daniel K. Shenfeld**[2]  **Ravi B. Parikh**[2,3]  **Jenna Wiens**[1]  **Maggie Makar**[1]
[1]University of Michigan  [2]University of Pennsylvania  [3]Emory University
{dylanz, mmakar}@umich.edu

## Abstract

In settings where ML models are used to inform the allocation of resources, agents affected by the allocation decisions might have an incentive to strategically change their features to secure better outcomes. While prior work has studied strategic responses broadly, disentangling misreporting from genuine adaptation remains a fundamental challenge. In this paper, we propose a causally-motivated approach to identify and quantify how much an agent misreports on average by distinguishing deceptive changes in their features from genuine adaptation. Our key insight is that, unlike genuine adaptation, misreported features do not causally affect downstream variables (i.e., causal descendants). We exploit this asymmetry by comparing the causal effect of misreported features on their causal descendants as derived from manipulated datasets against those from unmanipulated datasets. We formally prove identifiability of the misreporting rate and characterize the variance of our estimator. We empirically validate our theoretical results using a semi-synthetic and real Medicare dataset with misreported data, demonstrating that our approach can be employed to identify misreporting in real-world scenarios.

## 1 Introduction

Machine learning models are increasingly used by decision-makers to guide decisions about the allocation of critical resources, such as in loan applications, or determining government payouts to private insurers [2, 47, 14]. In these contexts, organizations—referred to as agents—may have an incentive to strategically change their features to secure better outcomes [32]. They can do so through genuine adaptation or misreporting. *Genuine adaptation* refers to agents genuinely changing their behavior, causing the actual values of their features to change. This leads to real improvements and authentic changes. Misreporting refers to agents not changing their behavior but instead reporting incorrect feature values to manipulate the allocation process, creating an illusion of improvement without any real change. Incentivizing genuine adaptation is often desirable to the decision-maker, as it can lead to improvements in the target outcome [41, 17]. Misreporting, however, is never desirable to the decision-maker as it leads to incorrect predictions and inefficient resource allocation.

As a running example, we consider the U.S. Medicare Advantage (MA) program, where the government uses a public risk adjustment model to allocate payments to privately-owned insurers based on enrollee health [37, 31]. This risk adjustment model is designed to recommend higher payments to insurers for higher-risk, sicker enrollees [3]. In response, private insurers may genuinely change their enrollees' health by providing targeted interventions to high-risk enrollees, reducing their costs and increasing their profit margins. Alternatively, they may resort to misreporting, or "upcoding," by inflating diagnosis codes to make enrollees appear sicker, which in turn increases payments without additional treatment expenses [23]. Upcoding has led to an estimated $50 billion in overpayments [29] and over $100 million in auditing efforts in 2024 alone [13]. Estimating the extent of misreporting is therefore critical for prioritizing oversight and promoting resource efficiency.

In this work, we develop a causal framework for detecting and quantifying misreporting in the presence of genuine adaptation. Our key insight is that misreporting, unlike genuine adaptation, does not causally affect the descendants of a given feature. Consequently, misreporting leads to biased causal effect estimates between the feature and its descendants. We exploit this asymmetry by comparing the estimated causal effect of a feature on its descendants in both manipulated and unmanipulated datasets to infer a misreporting rate. We assume access to both types of data, where the unmanipulated dataset is collected in a setting where there are no incentives to misreport, such as a setting prior to model deployment.

Our contributions are summarized as follows. (1) We recast the problem of quantifying the misreporting rate as a causal problem, showing that causal descendants of features can be used to distinguish changes due to genuine adaptation and misreporting. (2) We propose a novel estimator for the misreporting rate that leverages discrepancies in causal effect estimates computed from manipulated data and unmanipulated data where incentives to misreport are absent. (3) We provide theoretical guarantees, including conditions under which the misreporting rate is identifiable, and variance analysis of our estimator. (4) We empirically validate the performance of our estimator, showing that it outperforms baseline approaches on a semi-synthetic and real Medicare dataset. Code for our experiments is publicly available at https://github.com/DylanJamesZapzalka/misreporting_estimation.

## 2   Related Work

**Strategic Classification and Regression.** Our work is related to work on strategic classification and regression, where agents may change their features at some cost [15, 10, 28]. However, it differs in two key aspects. (1) The primary goal is to find a model that is robust to the distribution shifts caused by gaming, often relying on known agents' cost functions and iterative model retraining [36]. In contrast, we seek to estimate how much agents misreport. (2) Our method doesn't require the unrealistic assumptions of a known agent's cost function or iterative model training.

**Causal Strategic Classification.** Recent work views strategic classification/regression through a causal lens [32, 41, 17, 19] where agents can *only* genuinely adapt their features. They distinguish between two types of genuine adaptations: improvement and gaming, which correspond to modifications to features that are and are not causally related to the target label, respectively [32]. Unlike us, their focus is on creating models robust to both forms of genuine adaptation [19] and finding models that incentivize improvement over gaming [41, 17]. Closest to our work is Chang et al. [6], who propose an algorithm that can rank agents by their propensity to misreport their features. Unlike our work, their approach can only partially identify how much agents misreport, and they do not make a distinction between misreporting and genuine adaptation. Moreover, their approach works by estimating the causal effect of the agent on the observed feature, whereas our method leverages causal effect estimates on the causal descendants of the feature, which allows us to distinguish misreporting from genuine adaptation.

**Auditing Policies.** Other work seeks to disincentivize agents from misreporting their features through auditing [21, 11, 12]. They define a setting where the decision-maker deploys a transparent auditing policy that allows them to reveal the true features of a limited number of agents selected by the policy. If the agent's true features differ from their reported features, they endure a penalty, which incentivizes them to report their true features. Instead of performing costly audits, our work estimates misreporting by relying on additional unmanipulated data from settings where agents have no incentive to misreport, e.g., data collected before any model was deployed.

**Anomaly/Fraud Detection.** Our work is closely related to anomaly detection methods aimed at identifying fraudulent instances within a dataset, such as those arising in credit card transactions or insurance claims [18, 5]. Most relevant are one-class classification algorithms [40, 30, 39], which are trained on an unmanipulated dataset to detect anomalies in a manipulated dataset. Unlike our work, these methods focus on identifying specific instances that are anomalous or misreported, whereas our method estimates a rate of misreporting in a dataset. These methods also rely on the assumption that misreported data points differ significantly from normal data points. Our method instead relies on causal assumptions, specifically, that misreporting does not affect the causal descendants of features.

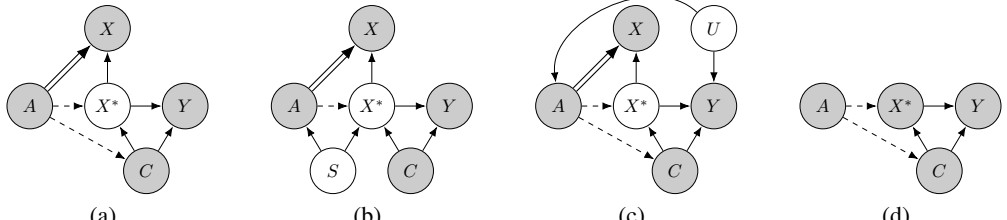

Figure 1: Causal DAGs that describe the setting of this paper. White nodes are unobserved whereas grey nodes are observed. Double-line arrows represent misreporting while dashed arrows represent genuine adaptation. DAGs (a)-(c) represent manipulated data-generating processes while DAG (d) represents unmanipulated data. The DAG in (a) represents our simplified main setting, (b) represents a setting with selection bias, (c) depicts a setting with unobserved confounding between $A$ and $Y$, and (d) represents unmanipulated data with potential genuine adaptation but no misreporting.

## 3 Preliminaries

**Setup.** We study a setting where some agents may either genuinely adapt and/or misreport their features. Let $A$ denote the agent identity, $X^*$ the true features, $X$ the (potentially misreported) agent-reported features, $Y$ a downstream variable causally influenced by $X^*$, and $C$ observed features that may act as confounders or effect modifiers for the relationship between $X^*$ and $Y$. We use uppercase letters to denote variables and lowercase letters for their realizations.

We assume that we have access to two datasets: (1) A possibly *manipulated dataset* $\mathcal{D} = \{(x_i, y_i, c_i, a_i)\}_{i=1}^N \sim P$, where $P$ follows any of the DAGs in 1(a)-(c). (2) An *unmanipulated dataset* $\mathcal{D}^* = \{(x_i^*, y_i, c_i)\}_{i=1}^M \sim P^*$ generated according to DAG 1(d). $\mathcal{D}^*$ may be pre-deployment data used to train the decision-making model, as agents have no incentive to manipulate their features before the model is deployed. We do not require access to agent identities in $\mathcal{D}^*$, nor do we assume that the same agents appear in both datasets.

In Figure 1, we use dashed arrows to indicate genuine adaptation and double-line arrows to indicate misreporting. In DAG 1(d), which represents the unmanipulated distribution, only genuine adaptations are allowed. For clarity, we present our main analysis assuming $\mathcal{D}$ is sampled according to the DAG in Figure 1(a). However, our results apply without modification to the more complicated DAGs in Figure 1(b)-(c), which include selection bias via an unobserved variable $S$ influencing both $A$ and $X^*$, or an unobserved confounder $U$ between $A$ and $Y$. Additional allowable DAGs are included in Appendix A.

**Running example.** In Medicare claims, $\mathcal{D}^*$ can be insurance claims covered directly by the government (i.e., Traditional Medicare (TM)), where there is no incentive to misreport. $\mathcal{D}$ can be claims covered by private insurers. Selection bias could happen, for example, when price-sensitive enrollees ($S$) who are less likely to have chronic conditions ($X^*$) prefer private insurance ($A$). Confounding between $A$ and $Y$ could occur if enrollees who prefer private insurance ($A$) tend to be sicker ($Y$).

**Assumptions.** We assume that $X^*$ and $X$ are binary whereas all other variables may be continuous. Without loss of generality, we assume that $X = 1$ is associated with a higher payout than $X = 0$ which means that agents are not incentivized to misreport features where $X^* = 1$, as formally stated in Assumption 1. We adopt the notation of the Neyman-Rubin potential outcomes framework [38], where $X(X^* = x^*)$ is defined as the counterfactual outcome that we would get if $X^*$ is set to $x^*$.

**Assumption 1** (Optimal Misreporting). $\forall i, \ x_i(x_i^* = 1) = 1$

We assume that agents are incentivized to misreport only $X^*$ and genuinely adapt it as follows.

**Assumption 2** (Useful Modifications). *Agents may only misreport $X^*$, or genuinely adapt it by intervening on $X^*$ or its ancestors.*

Under this setting, our goal is to determine how much an agent misreported their features, without access to the true features $X^*$. Letting $P_a(V) := P(V|A = a)$ for an arbitrary variable $V$, we define our estimand of interest as follows:

**Definition 1** (Misreporting Rate). $MR = P_a(X^* = 0 | X = 1)$

This definition of the MR quantifies the conditional probability that a reported feature is false. We also show in Appendix C that our analysis can be trivially extended to other variants such as the false positive rate, $P_a(X = 1 | X^* = 0)$, and the marginal difference $P_a(X = 1) - P_a(X^* = 1)$.

Key to our suggested approach will be our ability to estimate the causal effect of $X^*$ on $Y$. We also require the typical assumptions used in causal inference, defined below.

**Assumption 3.** *The features $X^*$, $C$, and the potential outcomes $Y(X^* = 1), Y(X^* = 0)$ satisfy the following properties:*

1. *No unmeasured confounding:* $Y(0), Y(1) \perp X^* \mid C$
2. *Overlap:* $P_a(X^* = x^* | C = c), P_a(X = x | C = c), P^*(X^* = x^* | C = c) > 0 \quad \forall x^*, x, c$
3. *Consistency:* $Y_i(x^*) = y_i$ *if* $X_i^* = x^*$.

Finally, we assume that the conditional average treatment effects are invariant across $P_a$ and $P^*$ for all values of $a$.

**Assumption 4.** $\mathbb{E}_{P_a}[Y(1) - Y(0) | C = c] = \mathbb{E}_{P^*}[Y(1) - Y(0) | C = c] \quad \forall c, a$

This assumption allows us to leverage unmanipulated data $\mathcal{D}^*$ to recover causal effects needed for estimating misreporting in $\mathcal{D}$.

## 4 Estimating Misreporting Rates

Recall that our goal is to estimate the misreporting rate for a given agent, defined in Definition 1. The core challenge lies in the fact that we only observe the reported features $X$, but not the true features $X^*$. This means that the estimand is not identifiable from $\mathcal{D}$ alone. A naive approach would be to combine $\mathcal{D}$ and $\mathcal{D}^*$ to estimate the causal effect of $A$ on the observed feature values. However, this would yield a biased estimate of the misreporting rate because it gives an estimate of *both* the effect of genuine adaptation and misreporting, i.e., a biased estimate of misreporting, as we establish in Appendix E. Instead, we estimate the misreporting rate by leveraging the distinct causal consequences that agent interventions corresponding to genuine adaptation and misreporting have on the downstream variables $Y$.

In Section 4.1, we present our main estimator for the misreporting rate. In Section 4.2, we analyze the variance of our estimator and give guidance on how to generate a low-variance estimator.

### 4.1 Our causal misreporting estimator (CMRE)

Our key insight is that genuine adaptation and misreporting have different effects on the causal descendants of $X^*$. When an agent genuinely adapts $X^*$, this results in a change to its causal descendant, $Y$. In contrast, when agents misreport, they only change the reported feature $X$, which doesn't causally affect $Y$. To illustrate, in our running Medicare example, if an enrollee truly has cancer ($X^* = 1$), this will cause them to experience cancer-related symptoms ($Y$). In contrast, if an insurer falsely reports a cancer diagnosis for an enrollee ($X^* = 0$, $X = 1$), the misreporting itself has no causal effect on the patient's symptoms.

This difference allows us to use the causal effect of $X$ on $Y$ and that of $X^*$ on $Y$ as a signature to distinguish between misreporting and genuine adaptation. To make progress, we define the "reported" group as the group for whom $X = 1$ and define the true average feature effect on the reported (TAFR) and the nominal average feature effect on the reported (NAFR) as follows:

$$\tau_a^* := \int_C (\mathbb{E}_{P_a}[Y(X^* = 1) | C = c] - \mathbb{E}_{P_a}[Y(X^* = 0) | C = c]) P_a(C = c | X = 1) dc,$$

and

$$\tau_a := \int_C (\mathbb{E}_{P_a}[Y | X = 1, C = c] - \mathbb{E}_{P_a}[Y | X = 0, C = c]) P_a(C = c | X = 1) dc,$$

respectively. These two expressions are similar to the average treatment effect on the treated, a commonly studied estimand in causal inference. They define the true and nominal causal effects for

the population for whom $X = 1$. We focus on causal effects for the "reported" group defined by $X = 1$ because the MR as defined in Defnition 1 is a conditional estimand over $X = 1$.

Although observing a difference in $\tau_a^*$ and $\tau_a$ indicates that an agent misreported, it doesn't give us a rate at which an agent misreports. To obtain the misreporting rate, we must compare the difference in the NAFR and TAFR relative to the baseline causal effect of $X^*$ on $Y$ for the group that is misreported. Since only the variable $X^*$ influences an agent's decision to misreport a datapoint, the misreported group will be a random sample of the group where $X^* = 0$ once we condition on the agent. Therefore, the average causal effect of $X^*$ on $Y$ for the misreported will be the average causal effect on the datapoints where $X^* = 0$. We define this as the true average feature effect on the misreported (TAFM):

$$\delta_a^* := \int_C (\mathbb{E}_{P_a}[Y(X^* = 1)|C = c] - \mathbb{E}_{P_a}[Y(X^* = 0)|C = c])P_a(C = c|X^* = 0)dc.$$

Next, we show that the MR can – in principle – be quantified as the rate of the differences between the TAFR and NAFR over the TAFM.

**Lemma 1** (Estimator for the misreporting rate). *Let Assumptions 1-3 hold. Then for $\delta_a^* \neq 0$, the MR can be expressed as:*

$$P_a(X^* = 0|X = 1) = \frac{\tau_a^* - \tau_a}{\delta_a^*}.$$

The proof for Lemma 1 and all other statements in this subsection are presented in Appendix B. The lemma states that we can quantify the MR by comparing the true and nominal causal effects of $X^*$ on $Y$ and $X$ on $Y$ respectively. While instructive, Lemma 1 is not very useful as we do not have access to $X^*$ for agent $a$, and hence we cannot directly estimate $\tau_a^*$ or $\delta_a^*$ from the manipulated data alone.

To resolve this issue, we leverage the unmanipulated dataset $\mathcal{D}^*$ to estimate two other quantities in place of $\tau_a^*$ and $\delta_a^*$. Specifically, we define

$$\tau_a' := \int_C (\mathbb{E}_{P^*}[Y(X^* = 1)|C = c] - \mathbb{E}_{P^*}[Y(X^* = 0)|C = c])P_a(C = c|X = 1)dc$$

and

$$\delta_a' := \int_C (\mathbb{E}_{P^*}[Y(X^* = 1)|C = c] - \mathbb{E}_{P^*}[Y(X^* = 0)|C = c])P_a(C = c|X = 0)dc.$$

Both $\tau_a'$ and $\delta_a'$ are identifiable because $X^*$ is observed in the unmanipulated dataset and can be used as valid estimators of $\tau_a^*$ and $\delta_a^*$ to estimate the MR, as we show in Theorem 1.

**Theorem 1** (Identifiability). *Let Assumptions 1-4 hold. Then for $\delta_a' \neq 0$, $P_a(X^* = 0|X = 1)$ is identifiable and can be expressed as:*

$$P_a(X^* = 0|X = 1) = \frac{\tau_a' - \tau_a}{\delta_a'}.$$

The proof of Theorem 1 relies on (1) the invariance of conditional causal effects of $X^*$ on $Y$ across $P_a$ and $P^*$ and (2) our assumptions about agent behavior to show that $\tau_a' = \tau_a^*$ and $\delta_a' = \delta_a^*$. We then show that the misreporting rate is identifiable as both $\delta_a'$ and $\tau_a'$ are identifiable from $\mathcal{D}, \mathcal{D}^*$, and standard causal effect assumptions.

**Approach.** Guided by Theorem 1, we now introduce our primary estimator – the causal misreporting estimator (CMRE) – which can estimate the misreporting rate for each agent. CMRE proceeds by estimating the conditional average treatment effect (CATE) over $\mathcal{D}^*$ and for an agent $a$ over $\mathcal{D}$, which we denote as $\theta^*(c)$ and $\theta_a(c)$ using typical causal estimators such as S-learners, T-learners [27], double/debiase ML methods [9, 33] and doubly robust estimators [25]. We demonstrate the estimation process using S-learners, which we used for our empirical analysis in Section 5. Specifically, for each agent $a$ in $\mathcal{D}$ we estimate the following

$$f_a(c, x) = \arg\min_{f \in \mathcal{F}} \frac{1}{N_a} \sum_{i : i \in \mathcal{D}, a_i = a} \ell(f(c_i, x_i), y_i), \quad \text{and} \quad \theta_a(c) := f_a(c, 1) - f_a(c, 0) \quad (1)$$

where $\mathcal{F}$ is a suitable function class, $\ell$ is some loss function and $N_a$ is the number of data points in $\mathcal{D}$ for which $A = a$. Similarly, we can get an estimate of $\theta^*(c)$:

$$f^*(c, x^*) = \arg\min_{f \in \mathcal{F}} \frac{1}{M} \sum_{i: i \in \mathcal{D}^*} \ell(f(c_i, x_i^*), y_i) \quad \text{and} \quad \theta^*(c) := f^*(c, 1) - f^*(c, 0). \quad (2)$$

Finally, our estimates for $\tau_a'$, $\tau_a$ and $\delta'$ can be computed as follows, where $N_{ax}$ is the number of datapoints in $\mathcal{D}$ where $A = a$ and $X = x$:

$$\hat{\tau}_a' = \frac{1}{N_{a1}} \sum_{\substack{i: i \in \mathcal{D}, x_i = 1, \\ a_i = a}} \theta^*(c_i), \quad \hat{\tau}_a = \frac{1}{N_{a1}} \sum_{\substack{i: i \in \mathcal{D}, x_i = 1, \\ a_i = a}} \theta(c_i), \quad \hat{\delta}_a' = \frac{1}{N_{a0}} \sum_{\substack{i: i \in \mathcal{D}, x_i = 0, \\ a_i = a}} \theta^*(c_i). \quad (3)$$

Our full algorithm is summarized in Appendix E. We note that our approach can be extended to estimate conditional versions of the misreporting rate by estimating CATE versions of the TAFR, NAFR, and TAFM estimands conditioned on any features of interest. Although this may increase the variance, it may be mitigated by using double ML methods [9] rather than S-learners.

## 4.2 Analyzing the variance of our estimator

In this section, we analyze the asymptotic variance of our estimator to give guidance as to how to obtain a low-variance estimator of the MR. We aim to show that if multiple causal descendants of $X^*$ are known, the causal descendent with the strongest causal relationship to $X^*$ should be used as it will result in the lowest asymptotic variance. For simplicity, we limit our analysis to a setting where the MR can be estimated from parametric estimators $\hat{\tau}_a$, $\hat{\tau}_a'$, and $\hat{\delta}_a'$ that are asymptotically normal. We characterize the asymptotic variance of a MR estimator under these assumptions in Theorem 2.

**Theorem 2** (Variance). *Let $\hat{\tau}_a$, $\hat{\tau}_a'$, and $\hat{\delta}_a'$ be asymptotically normal estimators with an asymptotic variance of $\sigma_{\tau_a}^2$, $\sigma_{\tau_a'}^2$, and $\sigma_{\delta_a'}^2$. Also let $\sigma_{\tau_a \tau_a'}$, $\sigma_{\tau_a \delta_a'}$, and $\sigma_{\delta_a' \tau_a'}$ denote the covariance between the estimators and $\xrightarrow{d}$ denote convergence in distribution. Suppose that $N = M = n$, then for $\delta_a' \neq 0$ and $\hat{\delta}_a' \neq 0$,*

$$\sqrt{n}\left[\frac{\hat{\tau}_a' - \hat{\tau}_a}{\hat{\delta}_a'} - \frac{\tau_a' - \tau_a}{\delta_a'}\right] \xrightarrow{d} \mathcal{N}\left(0, \frac{\sigma_{\tau_a'}^2 + \sigma_{\tau_a}^2 - 2\sigma_{\tau_a' \tau_a}}{\delta_a'^2} + 2\frac{\tau_a - \tau_a'}{\delta_a'^3}(\sigma_{\tau_a' \delta_a'} - \sigma_{\tau_a \delta_a'}) + \frac{(\tau_a - \tau_a')^2}{\delta_a'^4}\sigma_{\delta_a'}^2\right)$$

Theorem 2 shows that the asymptotic variance will increase significantly as $\delta_a' \to 0$ as each term in the variance is divided by either $\delta_a'^2$, $\delta_a'^3$, and $\delta_a'^4$. Thus, the causal descendent with the largest estimate of $\delta_a'$ should be used to calculate the MR.

## 5 Empirical Results

We evaluate the performance of our approach (CMRE) on semi-synthetic and real-world data. We show that CMRE consistently yields reliable estimates of the MR, even when genuine adaptation is present, and outperforms relevant baselines.

**Baseline Algorithms** We compare CMRE against the following baselines. (1) Natural Direct Effect Estimator (**NDEE**): this estimates the natural direct effect of the agent $A$ on the feature $X$ as a proxy for MR. We expect it to fail when genuine adaptation is non-zero, as it cannot distinguish between the direct causal effect of $A$ (misreporting) and the causal effect mediated through $X^*$ (genuine adaptation) without access to $X^*$. (2) Naive Misreporting Estimator (**NMRE**): this is similar to our approach but doesn't control for confounding between $X^*$ and $Y$. It is used to highlight the importance of controlling for confounding when estimating the MR. (3) One-Class SVM (**OC-SVM**): an anomaly detection approach trained on $\mathcal{D}^*$ where $X^* = 1$. It then detects which data points in $\mathcal{D}$ where where $X = 1$ are outliers. It highlights the limitations of using anomaly detection methods for MR estimation. Both CMRE and NDEE use S-learners [27] with an XGBoost [8] for causal effect estimation. NMRE relies on a difference-in-means estimator since it doesn't control for confounding. Additional implementation details are in Appendix D.

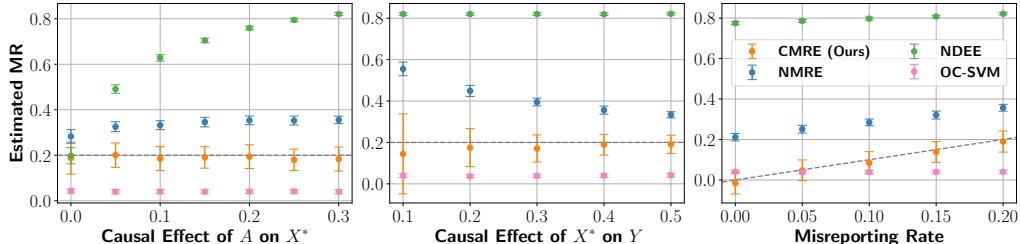

Figure 2: Results from the loan fraud dataset. The $x$-axis is the causal effect of $A$ on $X^*$ **(left)**, causal effect of $X^*$ on $Y$ **(middle)**, and the misreporting rate **(right)**. The $y$-axis is the estimated misreporting rate. Dashed lines represent the true misreporting rate and the error bars represent the standard deviation. Our approach (CMRE) accurately estimates the MR for all levels of genuine adaptation, the causal effect of $X^*$ on $Y$, and misreporting rates. The variance for our estimator depends on the magnitude of the causal effect of $X^*$ on $Y$. Baselines that do not adjust for confounding (NMRE) or do not distinguish between genuine adaptation and misreporting (NDEE) give biased estimates in various cases. Anomaly detection methods (OC-SVM) are unable to distinguish misreported data points from unmanipulated data points.

## 5.1 Semi-synthetic data experiments: Loan fraud

We simulate a scenario where loan applicants may misreport their employment status to improve their chances of loan approval. Here, we focus on a simple setting where we're interested in the overall MR rather than the MR for a specific agent. We leave the multi-agent analysis to Section 5.2.

We simulate a setting where $\mathcal{D}^*$ was collected before a model used to estimate the risk of default was deployed and $\mathcal{D}$ is the data collected after. We extract the confounders from a real credit card dataset ($n = 30,000$) [45, 46], with $C = (C_A, C_S, C_E, C_M)$ denoting age, sex, education, and marital status. We simulate the outcome $Y$, true employment status $X^*$, reported employment status $X$, and agent identity $A$ as follows:

$$A_i \sim \text{Bernoulli}(0.05 + 0.3(1 - C_{Si}) + 0.3(1 - C_{Mi})),$$
$$X_i^* \sim \text{Bernoulli}(0.05 + 0.05C_{Ei} + 0.3C_{Si}C_{Mi} + 0.1C_{Ai}^2 + \beta_A A_i),$$
$$Y_i \sim \text{Bernoulli}(0.05 + 0.05C_{Ei} + 0.3C_{Si}C_{Mi} + 0.1C_{Ai}^2 + \beta_{X^*}X_i^*),$$
$$X_i \sim X_i^* + A_i(1 - X_i^*)\text{Bernoulli}(\mu),$$

where $\mu$ is picked to target a desired MR (default = 0.2) and we set $\beta_A = 0.3$ and $\beta_{X^*} = 0.4$ unless otherwise specified. All samples with $A_i = 1$ are assigned to $\mathcal{D}$ and the rest to $\mathcal{D}^*$.

Each experiment is repeated 100 times each with a different draw of $A$, $X$, $X^*$, and $Y$. We use an 80/20 train/test split of $\mathcal{D}$ where our models are trained on the larger split and the MR is estimated using the smaller split. We do not split $\mathcal{D}^*$ as it is only used to train the models. We present the means and standard deviations of the MR estimates across these 100 runs. Additional results and details can be found in Appendix D.

**Varying the amount of genuine adaptation.** First, we examine how changes in genuine adaptation affect the MR estimates, highlighting the need to account for genuine adaptation when estimating the MR. To vary the amount of genuine adaptation, we simulate 7 settings with $\beta_A$ between 0.0 and 0.3.

The results are shown in Figure 2 (left), where the $x$-axis shows the different values of $\beta_A$ and the $y$-axis is the estimated MR. The dashed line shows the true value of the MR. The results show that our approach (CMRE) gives unbiased, stable estimates of the MR that are unaffected by the level of genuine adaptation. This signals that CMRE can successfully disentangle misreporting from genuine adaptation. By contrast, genuine adaptation affects the estimates from NMRE and NDEE. Notably, NMRE generally gives biased estimates because it does not control for confounding. As genuine adaptation increases, NMRE gives worse estimates due to a shift in the relationship between $X^*$ and $C$ across $\mathcal{D}$ and $\mathcal{D}^*$ that NMRE does not control for by not conditioning on $C$. Despite being rooted in mediation analysis, NDEE is unable to disentangle the direct causal effect of $A$ on $X$ from the

effect mediated through $X^*$ without access to $X^*$. This means that it gives an unbiased estimate *only* when genuine adaptation is zero, i.e., there is no path from $A$ to $X$ through $X^*$. OC-SVM gives biased estimates for all levels of genuine adaptation because the misreported data points are still plausible under $\mathcal{D}^*$.

**Varying the causal effect of $X^*$ on $Y$.** Next, we empirically test our characterization of the variance of our estimator (Theorem 2) by varying the causal effect of $X^*$ on $Y$. To vary the causal effect of $X^*$ on $Y$, we simulate 5 settings with the causal effect of $X^*$ on $Y$, $\beta_{X^*} = [0.1, 0.2, 0.3, 0.4, 0.5]$.

The results are shown in Figure 2 (middle), where the $x$-axis shows the causal effect of $X^*$ on $Y$, the $y$-axis represents the estimated MR, and the dashed line denotes the true MR. The results show that our approach (CMRE) is unbiased, regardless of what the causal effect between the feature and outcome is. However, small causal effects result in high-variance estimates for CMRE consistent with Theorem 2. The NMRE estimates are biased but do improve as the causal effect of $X^*$ on $Y$ increases, as the unobserved confounders have a diminished contribution towards the full estimate as the causal effect $X^*$ on $Y$ itself increases. In contrast, the estimates of NDEE do not vary, as the causal effect of $X^*$ on $Y$ has no impact on the causal effect of $A$ on $X$. Similarly, the estimates of OC-SVM are biased but invariant because the misreported data points are still plausible under $\mathcal{D}^*$.

**Varying the misreporting rate.** Finally, we consider how varying the true MR affects our approach and the baselines. To that end, we simulate 5 datasets with a MR between 0 and 0.2.

Figure 2 (right) shows the results. The $x$-axis and the dashed line show true MR whereas the $y$-axis shows the estimated MR. The results show that, unlike the baselines, our approach is always able to accurately estimate the MR. NDEE gives biased estimates since the genuine adaptation is non-zero. OC-SVM is invariant to the MR, indicating that it is completely unable to distinguish between normal and misreporting samples. The inaccuracy in NMRE arises due to uncontrolled confounding whereas by controlling for all confounders, CMRE leads to accurate estimates.

## 5.2 Real data experiments: Misreporting in Medicare insurance claims

Next, we highlight the utility of our approach in a real data setting. We aim to identify if private insurers misreport enrollees' diagnoses to secure higher government payouts, and which insurers have a higher misreporting rate.

**Background.** The government calculates payouts using a public risk adjustment model based on enrollee demographics (age, sex, race) and current diagnoses ($X^*$), as measured by Hierarchical Condition Categories (HCCs) at year ($t-1$) [37] to predict future healthcare costs. This model is trained on Traditional Medicare (TM) enrollees, who only use government insurance, which provides an unmanipulated dataset, $\mathcal{D}^*$, as there's no incentive to misreport. In contrast, data from private insurers, $\mathcal{D}$, may be manipulated. We expect to find evidence of misreporting or "upcoding" of HCCs included in the risk adjustment model by private insurers, consistent with existing literature [14, 42]. Estimating misreporting rates has been difficult, as private insurers claim diagnosis rate differences result from their improved care (i.e., genuine adaptation). For instance, private insurers may provide greater access to wellness benefits, screening assessments, and home-based care that result in an improvement of various conditions, while also identifying conditions that would have remained unreported in TM. Our proposed estimator can make progress toward resolving this long-standing issue by distinguishing between genuine adaptation and misreporting.

**Setup.** We use mortality during year $t$ as the downstream outcome $Y$ and consider the set of confounders to be enrollee demographics as well as HCCs measured in the previous year, $t-1$. To ensure that $C$ is unmanipulated, we consider only the population of "switchers": enrollees who were covered by the government in year $t-1$ but switched to a private insurer in year $t$. For these enrollees, their possible confounding HCCs ($C$) were reported during their government coverage but the HCCs used to calculate risk ($X$) were reported during their private coverage.

Here, we do not have access to the ground truth MR. Instead, we gauge the quality of our MR estimates for HCCs included in the payment model by comparing them to estimates of non-payment HCCs: diagnoses that are not included in the government's risk adjustment models. Private insurers have no incentive to misreport these HCCs and we expect the true MR for the non-payment HCCs to be zero. To obtain a 95% confidence interval for our estimator, we used bootstrapping with 1000 samples. To ensure a small interval, as represented by the error bars, we only considered the

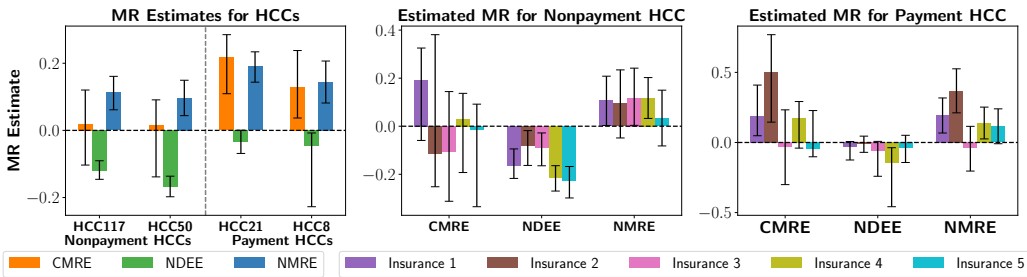

Figure 3: For each plot, the $y$-axis represents the estimated MR for an HCC code and the error bars represent a 95% confidence interval. **(Left)** The $x$-axis has two nonpayment HCCs (HCC117 and HCC50) and two payment HCCs (HCC21 and HCC8). Our approach (CMRE) has a MR estimate close to zero for nonpayment HCCs and significantly above zero for the payment HCCs, which aligns with what is expected in current literature. Baselines that fail to distinguish genuine adaptation from strategic adaptation (NDEE) seem to underestimate the MR and baselines that do not control for confounding (NMRE) seem to overestimate the MR. **(Middle and Right)** The $x$-axis represents the baselines. The middle plot represents estimates for HCC50 and the right plot represents MR estimates for HCC8 across different private insurers (agents). Similar to the left plot, NDEE seems to underestimate the MR across most agents and NMRE overestimates.

payment and nonpayment HCCs where at least 1% of switcher enrollees in year $t$ had the HCC code, and the causal effect of the HCC on $Y$, as estimated using $\mathcal{D}^*$, is greater than 0.1 (consistent with Theorem 2). To further reduce variance, future work can explore alternative causal descendants that exhibit stronger causal relationships with each feature or employ different causal effect estimators (e.g., doubly robust or double ML methods [25, 9]).

We conduct two sets of analysis: in the first, we present the overall misreporting rate for all insurance companies (i.e., all agents) on the top two payment and non-payment HCCs with the highest causal effect estimates on Y. Second, we look at variations in the MR across multiple agents, i.e., multiple insurance companies. Results for OC-SVM are only in the Appendix, as the method fails to distinguish between normal and misreported samples, consistent with observations in the semi-synthetic results. We include additional results and details in Appendix F and D.

**Results.** Figure 3 (left) shows the MR estimates for all insurance companies for two nonpayment HCCs (left of the vertical dashed line) and two payment HCCs (right of the vertical line). Our approach (CMRE) is the only approach that passes the sanity check: it gives MR estimates that are not statistically significantly different from zero for the nonpayment HCCs. This is consistent with our expectation that private insurers have no incentive to misreport nonpayment HCCs. Conversely, CMRE estimates significantly high misreporting rates for both of the payment HCCs. These results are validated in the health policy literature. For instance, HCC21 (Protein Calorie Malnutrition), has long been suspected to be improperly reported as it is reported at much higher rates than TM, leading it to be removed from the risk adjustment model years after our data was collected [4, 26]. In contrast, NMRE estimates a high misreporting rate for all HCCs, NDEE estimates a negative misreporting rate for all HCCs, which does not align with what is expected. All OC-SVM results are included only in Appendix F since they are consistently incorrect.

For the agent-level analysis, we focus on the top payment and the top nonpayment HCC to simplify our visualizations. Figure 3 (middle and right) show that the main conclusions from the aggregate level analysis hold on the agent-level: our approach is the only approach that reports MR rates not statistically significantly different from zero for nonpayment HCCs across all insurance companies. Our approach estimates that 2 out of the 5 insurance companies have misreporting rates that are statistically significantly different from zero. None of the baselines give consistently reliable estimates for the nonpayment HCCs. NDEE continues to underestimate the MR whereas NMRE overestimates the MR.

## 6 Conclusion

In this work, we propose a causal approach to estimating how much strategic agents misreport their features. We show that the misreporting rate is fully identifiable by comparing causal effect estimates

between a possibly manipulated and unmanipulated dataset. We also analyze the variance of our estimator, showing that a decision maker can accurately estimate the misreporting rate of a feature given a causal descendent with a strong causal relationship. We highlight the utility of our approach across empirical experiments over a semi-synthetic and a real Medicare dataset.

**Limitations, Broader Impacts, and Future Work.** Our work introduces a novel framework for disentangling genuine adaptation from misreporting, opening several promising directions for future research. First, our method assumes the absence of unobserved confounding, a standard but untestable assumption in causal inference. Future extensions could incorporate sensitivity analysis techniques to assess robustness under potential violations of this assumption [44, 24, 22]. Second, the current framework focuses on binary misreported features; extending it to categorical and continuous variables remains an important next step toward broader applicability. Finally, in many real-world settings, downstream causal variables may be unobserved or only observed after a long delay. Addressing this limitation through the use of surrogate or proxy outcomes [1] offers another valuable direction for future work.

We also acknowledge that this method could be used to disproportionately target certain organizations that may not have the resources to adequately respond to claims of misreporting. To mitigate this issue, we recommend the integration of impact assessments to be used alongside our method to mitigate such disparities.

## Acknowledgments and Disclosure of Funding

We thank the reviewers for their insightful comments. Special thanks to Ezekiel Emanual, Claudia Shi, Cecilia Ehrlichman, and Rohan Singh for their valuable feedback. We also thank Michael Shafir and the Advanced Research Computing team at the University of Michigan for assistance with data access and usage, as well as Will Ferrel for coordinating our meetings. This research was supported in part through computational resources and services provided by Advanced Research Computing at the University of Michigan, Ann Arbor. The authors are supported by a grant from Schmidt Futures (Award No. 70960). The funders had no role in the study design, analysis of results, decision to publish, or preparation of the manuscript. This study was deemed exempt and not regulated by the University of Michigan institutional review board (IRBMED; HUM00230364). This material is based upon work supported by the National Science Foundation Graduate Research Fellowship Program under Grant No. DGE-2241144. Any opinions, findings, and conclusions or recommendations expressed in this material are those of the author(s) and do not necessarily reflect the views of the National Science Foundation.

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

# A    Additional DAGs

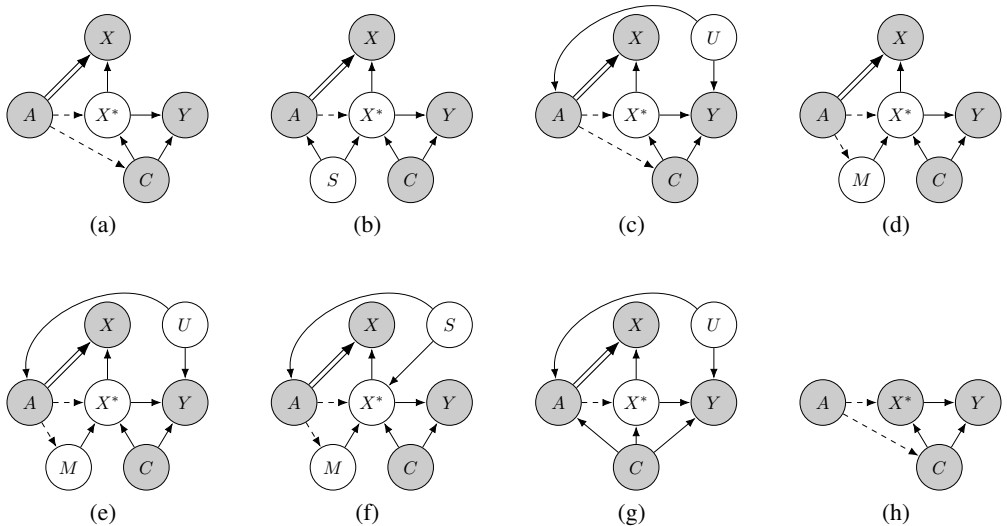

Figure 4: Causal DAGs that describe the setting of this paper. White nodes are unobserved, whereas grey nodes are observed. Double-line arrows represent misreporting, while dashed arrows represent genuine adaptation. DAGs (a)-(g) represent manipulated data-generating processes, while DAG (h) represents unmanipulated data.

Figures 4(a)-4(g) represent settings in which agents may either genuinely adapt or misreport their features. In contrast, Figure 4(h) represents a scenario involving trustworthy agents that only genuinely adapt their features. In all cases shown in Figures 4(a)-4(g), the decision maker lacks access to $X^*$ but observes $X$, $A$, $C$, and $Y$. While the main focus of the paper was on the DAG in Figure 4(a), our findings extend naturally to the more complex settings depicted in Figures 4(b)-4(g).

Specifically, the DAGs in Figures 4(b) and 4(f) represent scenarios where some unknown confounding bias may exist between $A$ and $X^*$, e.g., $S$. In the context of the Medicare example discussed in the main text, this could arise if enrollees with more chronic conditions ($X^*$) are more likely to enroll in a private health insurance plan ($A$). Notably, our approach doesn't require controlling for $S$, as it's not a confounder between $X^*$ and $Y$.

The DAGs in Figures 4(c), 4(e), and 4(g) illustrate settings where an unobserved confounder may influence both $A$ and $Y$. For example, this could occur if enrollees who prefer private insurance plans ($A$) also happen to have worse health outcomes ($Y$). Again, our approach does not require controlling for $U$. Although $U$ is a confounder of $X^*$ and $Y$, the backdoor path can be blocked by conditioning on $A$, which means that an adjustment for $U$ is unnecessary.

Finally, the DAGs in Figures 4(d)-4(f) capture settings where an agent may genuinely adapt their features by intervening on a mediator $M$ that lies between $A$ and $X^*$. For example, this could occur if private health insurers ($A$) are more likely to offer free gym memberships ($M$), which influence the true health status of their enrollees ($X^*$). As before, our approach does not require any knowledge of the mediators an agent intervenes on in order to estimate the misreporting rate, as $M$ is not a confounder of $X^*$ and $Y$.

# B   Main Proofs

Each of the proofs within this Section assume that the dataset $\mathcal{D} \sim P_a$ is generated according to any one of the DAGs in Figures 4(a)-4(g).

## B.1   Proof for Lemma 1

Lemma 1 is important to build up to Theorem 1. It shows that the MR can be estimated by comparing the true and nominal causal effects of $X^*$ on $Y$ and $X$ on $Y$.

**Lemma A1** (Estimator for the misreporting rate; Lemma 1 in the main text). *Let Assumptions 1-3 hold. Then for $\delta_a^* \neq 0$, the MR can be expressed as:*

$$P_a(X^* = 0 | X = 1) = \frac{\tau_a^* - \tau_a}{\delta_a^*}$$

*Proof.* Our proof proceeds in three main steps. First, we decompose $\tau_a$ into two terms: $\tau_a^*$ and an additional bias term. Second, we show that this additional term can be written as a function of our target estimand, $P(X^* = 0 | X = 1)$. Third and finally, we show that using simple algebra, we can express our target estimand as a function of $\tau_a^*$, $\tau_a$ and $\delta_a^*$

**Step 1: Decomposing $\tau_a$ into $\tau_a^*$ and an additional term**

$$\tau_a = \int_C (\mathbb{E}_{P_a}[Y|X=1,C=c] - \mathbb{E}_{P_a}[Y|X=0,C=c])P_a(C=c|X=1)dc$$

$$= \int_C \mathbb{E}_{P_a}[Y|X=1,C=c,X^*=1]P_a(X^*=1|X=1,C=c)P_a(C=c|X=1)dc$$

$$+ \int_C \mathbb{E}_{P_a}[Y|X=1,C=c,X^*=0]P_a(X^*=0|X=1,C=c)P_a(C=c|X=1)dc$$

$$- \int_C \mathbb{E}_{P_a}[Y|X=0,C=c]P_a(C=c|X=1)dc$$

$$= \int_C \mathbb{E}_{P_a}[Y|X^*=1,C=c](1-P_a(X^*=0|X=1,C=c))P_a(C=c|X=1)dc$$

$$+ \int_C \mathbb{E}_{P_a}[Y|X^*=0,C=c]P_a(X^*=0|X=1,C=c)P_a(C=c|X=1)dc$$

$$- \int_C \mathbb{E}_{P_a}[Y|X=0,C=c]P_a(C=c|X=1)dc$$

$$= \int_C \mathbb{E}_{P_a}[Y|X^*=1,C=c]P_a(C=c|X=1)dc$$

$$- \int_C \mathbb{E}_{P_a}[Y|X^*=1,C=c]P_a(X^*=0|X=1,C=c)P_a(C=c|X=1)dc$$

$$+ \int_C \mathbb{E}_{P_a}[Y|X^*=0,C=c]P_a(X^*=0|X=1,C=c)P_a(C=c|X=1)dc$$

$$- \int_C \mathbb{E}_{P_a}[Y|X^*=0,C=c]P_a(C=c|X=1)dc$$

$$= \tau_a^* - \int (\mathbb{E}_{P_a}[Y|X^*=1,C] - \mathbb{E}_{P_a}[Y|X^*=0,C])P_a(X^*=0|X=1,C)P_a(C|X=1)dc.$$

The third equality holds as $Y \perp X|X^*, A$ for the DAGs in Figure 4(a)-4(g) and the fourth equality holds due to Assumption 1.

**Step 2: Expressing the additional term as a function of** $P(X^*=0|X=1)$ Next, to explicitly show that this additional term is a direct consequence of misreporting, we can rewrite it in terms of the misrepoting rate:

$$\int_C (\mathbb{E}_{P_a}[Y|X^*=1,C] - \mathbb{E}_{P_a}[Y|X^*=0,C])P_a(X^*=0|X=1,C)P_a(C|X=1)dc$$

$$= \int_C (\mathbb{E}_{P_a}[Y|X^*=1,C] - \mathbb{E}_{P_a}[Y|X^*=0,C])\frac{P_a(X^*=0,C|X=1)}{P_a(C|X=1)}P_a(C|X=1)dc$$

$$= \int_C (\mathbb{E}_{P_a}[Y|X^*=1,C] - \mathbb{E}_{P_a}[Y|X^*=0,C])P_a(X^*=0|X=1)P_a(C|X^*=0,X=1)dc$$

$$= P_a(X^*=0|X=1)\int_C (\mathbb{E}_{P_a}[Y|X^*=1,C] - \mathbb{E}_{P_a}[Y|X^*=0,C])P_a(C|X^*=0,X=1)dc$$

$$= P_a(X^*=0|X=1)\int_C \mathbb{E}_{P_a}[Y|X^*=1,C] - \mathbb{E}_{P_a}[Y|X^*=0,C]P_a(C|X^*=0)dc$$

$$= P_a(X^*=0|X=1)\int_C \mathbb{E}_{P_a}[Y(X^*=1)|C] - \mathbb{E}_{P_a}[Y(X^*=0)|C=c]P_a(C|X^*=0)dc$$

$$= P_a(X^*=0|X=1)\delta_a^*.$$

The fourth equality comes directly from the fact that $C \perp X|X^*, A$ for the DAGs in Figures 4(a)-4(g). The fifth equality comes from Assumption 3, as all confounders are controlled for. Notably, both $M$ and $S$ are not confounders of $X^*$ and $Y$. The variable $U$ is a confounder of $X^*$ and $Y$, however, the

backdoor path is blocked by $A$, so it doesn't need to be directly controlled for. Overall, this shows that any difference between $\tau_a$ and $\tau_a^*$ is directly related to the misreporting rate.

**Step 3: Getting the expression for the final target estimand**    Finally, we can obtain a way to estimate the misreporting rate by rearranging the terms:

$$\tau_a = \tau_a^* - P_a(X^* = 0|X = 1)\delta_a^* \implies P_a(X^* = 0|X = 1) = \frac{\tau_a^* - \tau_a}{\delta_a^*},$$

for $\delta_a^* \neq 0$. Therefore, by comparing the difference in causal effects, we can identify the misreporting rate. $\square$

### B.2  Proof for Theorem 1

We now build upon the result from Lemma 1 as we work toward our main theorem. Before presenting the proof of Theorem 1, we first introduce an additional Lemma which shows that $\tau_a$, $\tau_a'$, and $\delta_a'$ are identifiable using $\mathcal{D}$ and $\mathcal{D}^*$, along with standard causal estimation assumptions. Then, in Theorem 1, we demonstrate that the misreporting rate is identifiable by showing that $\tau_a' = \tau_a^*$ and $\delta_a' = \delta_a^*$. This proof follows from Assumption 4, which states that the conditional causal effect of $X^*$ on $Y$ will remain invariant across both strategic and non-strategic populations.

**Lemma A2** (Identifiability of $\tau_a$, $\tau_a'$, and $\delta_a'$)**.** *Let Assumption 3 hold. Then $\tau_a$, $\tau_a'$, and $\delta_a'$ are identifiable using $\mathcal{D}$ and $\mathcal{D}'$.*

*Proof.*  First, recall that

$$\tau_a := \int_C (\mathbb{E}_{P_a}[Y|X = 1, C = c] - \mathbb{E}_{P_a}[Y|X = 0, C = c])P_a(C = c|X = 1)dc.$$

We know that $\tau_a$ is identifiable using only $\mathcal{D}$ as $Y$, $X$, and $C$ are all known in $\mathcal{D}$.

Next, recall that

$$\tau_a' := \int_C (\mathbb{E}_{P^*}[Y(X^* = 1)|C = c] - \mathbb{E}_{P^*}[Y(X^* = 0)|C = c])P_a(C = c|X = 1)dc$$

and

$$\delta_a' := \int_C (\mathbb{E}_{P^*}[Y(X^* = 1)|C = c] - \mathbb{E}_{P^*}[Y(X^* = 0)|C = c])P_a(C = c|X = 0)dc.$$

Again, we know that $P_a(C = c|X = 1)$ and $P_a(C = c|X = 0)$ are identifiable using only $\mathcal{D}$. Therefore, to show that $\tau_a'$ and $\delta_a'$ are identifiable, we must show that

$$\mathbb{E}_{P^*}[Y(X^* = 1)|C = c] - \mathbb{E}_{P^*}[Y(X^* = 0)|C = c]$$

is identifiable. This follows immediately from Assumptions 3:

$$\begin{aligned}
&\mathbb{E}_{P^*}[Y(X^* = 1)|C = c] - \mathbb{E}_{P^*}[Y(X^* = 0)|C = c]\\
&= \mathbb{E}_{P^*}[Y(X^* = 1)|X^* = 1, C = c] - \mathbb{E}_{P^*}[Y(X^* = 0)|X^* = 0, C = c]\\
&= \mathbb{E}_{P^*}[Y|X^* = 1, C = c] - \mathbb{E}_{P^*}[Y|X^* = 0, C = c]
\end{aligned}$$

Therefore, $\tau_a$, $\tau_a'$, and $\delta_a'$ are identifiable using $\mathcal{D}$ and $\mathcal{D}'$. $\square$

**Theorem A1** (Identifiability; Theorem 1 in the main text)**.** *Let Assumptions 1-4 hold. Then for $\delta_a' \neq 0$, $P_a(X^* = 0|X = 1)$ is identifiable and can be expressed as:*

$$P_a(X^* = 0|X = 1) = \frac{\tau_a' - \tau_a}{\delta_a'}.$$

*Proof.*  We know that $\tau_a$, $\tau_a'$, and $\delta_a'$ are identifiable using $\mathcal{D}$ and $\mathcal{D}^*$ by Lemma A2. Therefore, to complete this proof, we only need to show that $\tau_a' = \tau_a^*$ and $\delta_a' = \delta_a^*$, as implied by Lemma 1.

First, we show that $\tau'_a = \tau^*_a$. Recall that

$$\tau'_a := \int_C (\mathbb{E}_{P^*}[Y(X^* = 1)|C = c] - \mathbb{E}_{P^*}[Y(X^* = 0)|C = c])P_a(C = c|X = 1)dc$$

and

$$\tau^*_a := \int_C (\mathbb{E}_{P_a}[Y(X^* = 1)|C = c] - \mathbb{E}_{P_a}[Y(X^* = 0)|C = c])P_a(C = c|X = 1)dc.$$

Since

$$\mathbb{E}_{P_a}[Y(1) - Y(0)|C = c] = \mathbb{E}_{P^*}[Y(1) - Y(0)|C = c]$$

for all $c$ by Assumption 4, it follows immediately that $\tau'_a = \tau^*_a$.

Next, recall that

$$\delta'_a := \int_C (\mathbb{E}_{P^*}[Y(X^* = 1)|C = c] - \mathbb{E}_{P^*}[Y(X^* = 0)|C = c])P_a(C = c|X = 0)dc$$

and

$$\delta^*_a := \int_C (\mathbb{E}_{P_a}[Y(X^* = 1)|C = c] - \mathbb{E}_{P_a}[Y(X^* = 0)|C = c])P_a(C = c|X^* = 0)dc.$$

We already know that the conditional causal effects of $X^*$ on $Y$ are the same across $P^*$ and $P_a$. It remains to show that $P_a(C = c|X^* = 0) = P_a(C = c|X = 0)$ for all values of $c$ to show that $\delta'_a = \delta^*_a$. We establish this equality next.

To show this, we simply apply the law of total probability as follows:

$$\begin{aligned}
P_a(C = c|X = 0) &= P_a(C = c|X = 0, X^* = 1)P(X^* = 1|X = 0) \\
&\quad + P_a(C = c|X = 0, X^* = 0)P(X^* = 0|X = 0) \\
&= P_a(C = c|X = 0, X^* = 0) \\
&= P_a(C = c|X^* = 0).
\end{aligned}$$

The second equality follows because $P_a(X^* = 1|X = 0) = 0$ and $P(X^* = 0|X = 0) = 1$ by Assumption 1. The third equality follows as $C \perp X|A, X^*$ for all DAGs in Figures 4(a)-4(g). Note that this finding is intuitive: it can be traced back to the assumption that the agents pick who to misreport at random, which is implied by the DAGs.

Thus, the MR is identifiable.

$\square$

## B.3 Proof for Theorem 2

**Theorem A2** (Variance; Theorem 2 in the main text). *Let $\hat{\tau}_a$, $\hat{\tau}'_a$, and $\hat{\delta}'_a$ be asymptotically normal estimators with an asymptotic variance of $\sigma^2_{\tau_a}$, $\sigma^2_{\tau'_a}$, and $\sigma^2_{\delta'_a}$. Also let $\sigma_{\tau_a\tau'_a}$, $\sigma_{\tau_a\delta'_a}$, and $\sigma_{\delta'_a\tau'_a}$ denote the covariance between the estimators and $\xrightarrow{d}$ denote convergence in distribution. Suppose that $N = M = n$, then for $\delta'_a \neq 0$ and $\hat{\delta}'_a \neq 0$,*

$$\sqrt{n}[\frac{\hat{\tau}'_a - \hat{\tau}_a}{\hat{\delta}'_a} - \frac{\tau'_a - \tau_a}{\delta'_a}] \xrightarrow{d} \mathcal{N}(0, \frac{\sigma^2_{\tau'_a} + \sigma^2_{\tau_a} - 2\sigma_{\tau'_a\tau_a}}{\delta'^2_a} + 2\frac{\tau_a - \tau'_a}{\delta'^3_a}(\sigma_{\tau'_a\delta'_a} - \sigma_{\tau_a\delta'_a}) + \frac{(\tau_a - \tau'_a)^2}{\delta'^4_a}\sigma^2_{\delta'_a})$$

*Proof.* By the definition of asymptotic normality, each estimator has the following asymptotic distributions, where $\sigma^2_{\tau'_a}$ is asymptotic variance of $\hat{\tau}'_a$, $\sigma^2_{\tau_a}$ is asymptotic variance of $\hat{\tau}_a$, and $\sigma^2_{\delta'_a}$ is asymptotic variance of $\hat{\delta}'_a$:

$$\sqrt{n}[\hat{\tau}'_a - \tau'_a] \xrightarrow{d} \mathcal{N}(0, \sigma^2_{\tau'_a}),$$

$$\sqrt{n}[\hat{\tau}_a - \tau_a] \xrightarrow{d} \mathcal{N}(0, \sigma^2_{\tau_a}), \text{ and}$$

$$\sqrt{n}[\hat{\delta}'_a - \delta'_a] \xrightarrow{d} \mathcal{N}(0, \sigma^2_{\delta'_a}).$$

To proceed, we define the function $g(\hat{\tau}'_a, \hat{\tau}_a, \hat{\delta}'_a)$ as an estimator for the misreporting rate:

$$g(\hat{\tau}'_a, \hat{\tau}_a, \hat{\delta}'_a) = \frac{\hat{\tau}'_a - \hat{\tau}_a}{\hat{\delta}'_a}.$$

Since $\hat{\tau}'_a$, $\hat{\tau}_a$, $\hat{\delta}'_a$ are asymptotically normal, we can apply the delta method [43] to find the asymptotic variance of $g(\hat{\tau}'_a, \hat{\tau}_a, \hat{\delta}'_a)$, which states that

$$\sqrt{n}[g(\hat{\tau}'_a, \hat{\tau}_a, \hat{\delta}'_a) - g(\tau'_a, \tau_a, \delta'_a)] \xrightarrow{d} \mathcal{N}(0, \nabla g(\tau'_a, \tau_a, \delta'_a) \Sigma \nabla g(\tau'_a, \tau_a, \delta'_a)^\top),$$

where

$$\nabla g(\tau'_a, \tau_a, \delta'_a) = \begin{pmatrix} \frac{1}{\delta'_a} & \frac{-1}{\delta'_a} & \frac{\tau_a - \tau'_a}{\delta'_a{}^2} \end{pmatrix}$$

and

$$\Sigma = \begin{pmatrix} \sigma^2_{\tau'_a} & \sigma_{\tau_a \tau'_a} & \sigma_{\delta'_a \tau'_a} \\ \sigma_{\tau'_a \tau_a} & \sigma^2_{\tau_a} & \sigma_{\delta'_a \tau_a} \\ \sigma_{\tau'_a \delta'_a} & \sigma_{\tau_a \delta'_a} & \sigma^2_{\delta'_a} \end{pmatrix}$$

Therefore, we can calculate the asymptotic variance as follows:

$$\nabla g(\tau'_a, \tau_a, \delta'_a)^\top \Sigma \nabla g(\tau'_a, \tau_a, \delta'_a) = \begin{pmatrix} \frac{1}{\delta'_a} & \frac{-1}{\delta'_a} & \frac{\tau_a - \tau'_a}{\delta'_a{}^2} \end{pmatrix} \begin{pmatrix} \sigma^2_{\tau'_a} & \sigma_{\tau_a \tau'_a} & \sigma_{\delta'_a \tau'_a} \\ \sigma_{\tau'_a \tau_a} & \sigma^2_{\tau_a} & \sigma_{\delta'_a \tau_a} \\ \sigma_{\tau'_a \delta'_a} & \sigma_{\tau_a \delta'_a} & \sigma^2_{\delta'_a} \end{pmatrix} \begin{pmatrix} \frac{1}{\delta'_a} \\ \frac{-1}{\delta'_a} \\ \frac{\tau_a - \tau'_a}{\delta'_a{}^2} \end{pmatrix}$$

$$= \begin{pmatrix} \sigma^2_{\tau'_a} \frac{1}{\delta'_a} - \sigma_{\tau'_a \tau_a} \frac{1}{\delta'_a} + \sigma_{\tau'_a \delta'_a} \left( \frac{\tau_a - \tau'_a}{\delta'_a{}^2} \right) \\ \sigma_{\tau_a \tau'_a} \frac{1}{\delta'_a} - \sigma^2_{\tau_a} \frac{1}{\delta'_a} + \sigma_{\tau_a \delta'_a} \left( \frac{\tau_a - \tau'_a}{\delta'_a{}^2} \right) \\ \sigma_{\delta'_a \tau'_a} \frac{1}{\delta'_a} - \sigma_{\delta'_a \tau_a} \frac{1}{\delta'_a} + \sigma^2_{\delta'_a} \left( \frac{\tau_a - \tau'_a}{\delta'_a{}^2} \right) \end{pmatrix} \begin{pmatrix} \frac{1}{\delta'_a} \\ \frac{-1}{\delta'_a} \\ \frac{\tau_a - \tau'_a}{\delta'_a{}^2} \end{pmatrix}$$

$$= \sigma^2_{\tau'_a} \frac{1}{\delta'_a{}^2} - \sigma_{\tau'_a \tau_a} \frac{1}{\delta'_a{}^2} + \sigma_{\tau'_a \delta'_a} \frac{\tau_a - \tau'_a}{\delta'_a{}^3}$$

$$- \sigma_{\tau_a \tau'_a} \frac{1}{\delta'_a{}^2} + \sigma^2_{\tau_a} \frac{1}{\delta'_a{}^2} - \sigma_{\tau_a \delta'_a} \frac{\tau_a - \tau'_a}{\delta'_a{}^3}$$

$$+ \sigma_{\delta'_a, \tau'_a} \frac{\tau_a - \tau'_a}{\delta'_a{}^3} - \sigma_{\delta'_a \tau_a} \frac{\tau_a - \tau'_a}{\delta'_a{}^3} + \sigma^2_{\delta'_a} \frac{(\tau_a - \tau'_a)^2}{\delta'_a{}^4}$$

$$= \sigma^2_{\tau'_a} \frac{1}{\delta'_a{}^2} + \sigma^2_{\tau_a} \frac{1}{\delta'_a{}^2} - 2\sigma_{\tau'_a \tau_a} \frac{1}{\delta'_a{}^2}$$

$$+ 2\sigma_{\tau'_a \delta'_a} \frac{\tau_a - \tau'_a}{\delta'_a{}^3} - 2\sigma_{\tau_a \delta'_a} \frac{\tau_a - \tau'_a}{\delta'_a{}^3}$$

$$+ \sigma^2_{\delta'_a} \frac{(\tau_a - \tau'_a)^2}{\delta'_a{}^4}$$

$$= \frac{1}{\delta'_a{}^2} (\sigma^2_{\tau'_a} + \sigma^2_{\tau_a} - 2\sigma_{\tau'_a \tau_a})$$

$$+ 2\frac{\tau_a - \tau'_a}{\delta'_a{}^3} (\sigma_{\tau'_a \delta'_a} - \sigma_{\tau_a \delta'_a})$$

$$+ \frac{(\tau_a - \tau'_a)^2}{\delta'_a{}^4} \sigma^2_{\delta'_a}.$$

Therefore, $\sqrt{n}[\frac{\hat{\tau}'_a - \hat{\tau}_a}{\hat{\delta}'_a} - \frac{\tau'_a - \tau_a}{\delta'_a}]$ asymptotically converges to the following normal distribution:

$$\sqrt{n}[\frac{\hat{\tau}'_a - \hat{\tau}_a}{\hat{\delta}'_a} - \frac{\tau'_a - \tau_a}{\delta'_a}] \xrightarrow{d} \mathcal{N}(0, \frac{\sigma^2_{\tau'_a} + \sigma^2_{\tau_a} - 2\sigma_{\tau'_a \tau_a}}{\delta'_a{}^2} + 2\frac{\tau_a - \tau'_a}{\delta'_a{}^3} (\sigma_{\tau'_a \delta'_a} - \sigma_{\tau_a \delta'_a}) + \frac{(\tau_a - \tau'_a)^2}{\delta'_a{}^4} \sigma^2_{\delta'_a})$$

$$\square$$

## C   Additional Estimands

In this section, we show that if we can identify the main estimand of interest, $P_a(X = 1|X^* = 0)$, we can also identify other useful estimands, which are defined below.

**Definition 2** (Difference in Marginals). $DIM = P_a(X = 1) - P_a(X^* = 1)$.

**Definition 3** (False Positive Rate). $FPR = P_a(X = 1|X^* = 0)$.

The estimand in definition 3 can simply be interepreted as the false positive rate whereas the estimand in definition 2 can be thought of as the probability that a feature was misreported.

To establish that the estimand in definition 2 is identifiable, we first establish that our estimand of interest, $P_a(X = 1) - P_a(X^* = 0)$, can be expressed as the joint distribution $P_a(X = 1, X^* = 0)$ in Lemma A3. Identifiability follows from Theorem 1 and a simple application of Bayes rule as both $P_a(X^* = 0|X = 1)$ and $P_a(X = 1)$ are identifiable.

Additionally, since Lemma A3 implies that both $P_a(X = 1, X^* = 0)$ and $P_a(X^* = 0)$ are identifiable, we can show that the estimand in definition 3 is also identifiable.

**Lemma A3.** *Let Assumption 1 hold. Then* $P_a(X = 1) - P_a(X^* = 1) = P_a(X = 1, X^* = 0)$

*Proof.*

$$
\begin{aligned}
P_a(X = 1, X^* = 0) &= P_a(X = 1, X^* = 0) + P_a(X^* = 1) - P_a(X^* = 1) \\
&= P_a(X = 1, X^* = 0) + P_a(X = 1|X^* = 1)P_a(X^* = 1) - P_a(X^* = 1) \\
&= P_a(X = 1, X^* = 0) + P_a(X = 1, X^* = 1) - P_a(X^* = 1) \\
&= P_a(X = 1) - P_a(X^* = 1),
\end{aligned}
$$

where the second equality follows because $P_a(X = 1|X^* = 1) = 1$ by Assumption 1. $\square$

**Corollary A1** (Identifiability of difference in marginals). *Let Assumptions 1-4 hold. Then for* $\delta'_a \neq 0$, $P_a(X = 1) - P_a(X^* = 1)$ *is identifiable and can be expressed as:*

$$
P_a(X = 1) - P_a(X^* = 1) = \frac{\tau'_a - \tau_a}{\delta'_a} \times P_a(X = 1).
$$

*Proof.* The proof relys on a simple application of Bayes rule, and results from Lemma A3 and Theorem 1. Specifically, we have that:

$$
\begin{aligned}
P_a(X = 1) - P_a(X^* = 1) &= P_a(X = 1, X^* = 0) \\
&= P_a(X^* = 0|X = 1)P_a(X = 1),
\end{aligned}
$$

where the first equality follows by Lemma A3 and the second equality follows by Bayes rule. By theorem 1, the first term ($P_a(X^* = 0|X = 1)$) is identifiable, and $P_a(X = 1)$ is identifiable because all variables required for estimation are observed. $\square$

**Corollary A2** (Identifiability of false positive rate). *Let Assumptions 1-4 hold. Then for* $\delta'_a \neq 0$, $P_a(X = 1|X^* = 0)$ *is identifiable and can be expressed as:*

$$
P_a(X = 1|X^* = 0) = \frac{\tau'_a - \tau_a}{\delta'_a} \times P_a(X = 1).
$$

*Proof.* From Lemma A3, we can derive $P(X^* = 0)$ as follows:

$$
\begin{aligned}
P_a(X = 1) - P_a(X^* = 1) = P_a(X = 1, X^* = 0) &\implies \\
P_a(X = 1) - P_a(X = 1, X^* = 0) = P_a(X^* = 1) &\implies \\
1 - \{P_a(X = 1) - P_a(X = 1, X^* = 0)\} = 1 - P_a(X^* = 1) &\implies \\
P_a(X = 0) + P_a(X = 1, X^* = 0) = P_a(X^* = 0) &
\end{aligned}
$$

By Bayes' theorem, we can write the estimand as

$$P_a(X = 1 | X^* = 0) = \frac{P_a(X^* = 0 | X = 1) P_a(X = 1)}{P_a(X^* = 0)}$$
$$= \frac{P_a(X^* = 0 | X = 1) P_a(X = 1)}{P_a(X = 0) + P_a(X = 1, X^* = 0)}$$

Thus, since $P_a(X^* = 0 | X = 1)$, $P_a(X = 1, X^* = 0)$, $P_a(X = 1)$, and $P_a(X = 0)$ are identifiable, $P_a(X = 1 | X^* = 0)$ must be identifiable.

$\square$

## D   Datasets

### D.1   Medicare Dataset

The medicare dataset used in our experiments consists of insurance claims data from real U.S. Medicare enrollees enrolled in either Traditional Medicare or a private medicare insurance plan. The data was provided to the authors under a data usage agreement with the Centers for Medicare and Medicaid Services (CMS). For our experiments, we only use enrollees that had Medicare coverage in both 2019 ($t$) and 2018 ($t - 1$). We exclude enrollees who were dual-eligible (i.e., are eligible for both U.S. Medicaid and Medicare), had end-stage renal disease, or were below the age of 65 for the year $t - 1$. In addition, we exclude all enrollees who resided outside of the 50 U.S. states, the District of Columbia, Puerto Rico, or the U.S. Virgin Islands.

Each of the private medicare insurers is treated as a strategic agent that may misreport enrollee features. We used five agents in total for our experiments. Four agents correspond to the largest private insurers based on the total number of enrollees in year $t$. The fifth agent is created by aggregating the enrollees from all other smaller insurers. In contrast, Traditional Medicare was treated as a trustworthy agent that doesn't manipulate enrollee data, as there is no incentive to misreport.

The goal of our analysis is to assess how much private medicare insurers misreport HCC codes, which are binary variables that indicate if an enrollee has been diagnosed with a specific medical condition. We use V23 HCC codes, as defined by CMS, which are derived by mapping ICD-10 diagnosis codes reported in the claims data. There are two types of HCC codes: payment HCCs, which are used by a risk-adjustment model to predict future healthcare costs, and nonpayment HCCs, which are not used to determine costs. We expect the misreporting rate for each nonpayment HCC to be zero as there is no incentive for private insurers to misreport them.

For our analysis, we partition the enrollees into two different cohorts: stayers and switchers. To derive the stayers cohort, we sampled enrollees enrolled in Traditional Medicare for all 12 months in year $t - 1$ and were not enrolled in a private insurance plan in year $t$. For the switchers cohort, we used enrollees that were enrolled in Traditional Medicare for all 12 months in year $t - 1$ and were enrolled in a private insurance plan for at least one month in year $t$. We only used a 20% random sample of the eligible stayers cohort (868255 samples) and 100% of the eligible switchers cohort (166539 samples). For the outcome ($Y$), we use the enrollee's death status in year $t$.

For the features ($X$), we used both payment and nonpayment HCC codes, consisting of 83 and 99 codes, respectively. As covariates, we used the enrollee's age, race, sex, and the payment HCCs from year $t - 1$ to ensure they were not misreported. To obtain low variance estimates, we restricted our analysis to payment and nonpayment HCCs with the largest causal effects on death and where at least 1% of the switchers enrollees in year $t$ had the HCC code.

### D.2   Loan Datasets

In our loan dataset simulations, we model a setting where loan applicants may either genuinely adapt or misreport their employment status to improve their chances of getting approved for a loan. For each of our simulations, we simulate a single strategic agent ($A = 1$) and a single nonstrategic agent ($A = 0$). In addition to the semi-synthetic dataset used for the experiments in Section 5, we generate

additional datasets based on different DAGs in Figure 4. The data generation process for the other datasets is explained in Appendix F.

All of the simulations use the covariates extracted from a real credit card dataset [45]. These include three binary variables: marriage status ($C_M$), sex ($C_S$), and education ($C_E$), as well as another variable representing a person's age ($C_A$). We use min-max normalization so that $C_A$ is between 0 and 1. The agent variable ($A$), the variable for employment status ($X^*$), and the variable indicating if a loan applicant defaulted ($Y$), are all generated using the covariates. Misreporting is done in accordance with the following equation:

$$X_i \sim X_i^* + A_i(1 - X_i^*)\text{Bernoulli}(\mu)$$

where $\mu$ is picked to target a desired MR (default = 0.2). Each experiment is repeated 100 times, with new draws of $A, X, X^*$, and $Y$. Across all experiments, we use an 80/20 train/test split of $\mathcal{D}$.

# E   Estimators

In this section, we present additional details about our primary method (CMRE) as well as the baseline approaches (NMRE, NDEE, and OC-SVM). We also specify the hyperparameters and libraries used to implement each method in our experiments.

## E.1   CMRE

Recall that for a suitable function class $\mathcal{F}$, a loss function $\ell$, and $N_a$ – the number of data points in $\mathcal{D}$ for which $A = a$ – we define

$$f_a(c, x) = \arg\min_{f \in \mathcal{F}} \frac{1}{N_a} \sum_{i:i \in \mathcal{D}, a_i = a} \ell(f(c_i, x_i), y_i), \quad \text{and} \quad \theta_a(c) := f_a(c, 1) - f_a(c, 0) \quad (4)$$

and

$$f^*(c, x^*) = \arg\min_{f \in \mathcal{F}} \frac{1}{M} \sum_{i:i \in \mathcal{D}^*} \ell(f(c_i, x_i^*), y_i) \quad \text{and} \quad \theta^*(c) := f^*(c, 1) - f^*(c, 0). \quad (5)$$

Recall that $N_{ax}$ denotes the number of data points in $\mathcal{D}$ for which $A = a$ and $X = x$. Using this, we compute the estimates for $\tau_a', \tau_a$ and $\delta_a'$ as follows:

$$\hat{\tau}_a' = \frac{1}{N_{a1}} \sum_{\substack{i:i \in \mathcal{D}, x_i = 1, \\ a_i = a}} \theta^*(c_i), \quad \hat{\tau}_a = \frac{1}{N_{a1}} \sum_{\substack{i:i \in \mathcal{D}, x_i = 1, \\ a_i = a}} \theta(c_i), \quad \hat{\delta}_a' = \frac{1}{N_{a0}} \sum_{\substack{i:i \in \mathcal{D}, x_i = 0, \\ a_i = a}} \theta^*(c_i). \quad (6)$$

To estimate $\theta_a(c)$ and $\theta^*(c)$, we employ an S-learner, where the models $f_a$ and $f^*$ are implemented using XGBoost. We use the default hyperparameters provided by the XGBoost library in Python to train each model [7], including a learning rate of 0.3, a maximume tree depth of 6, and L2 regularization with a coefficient of 1.

The complete algorithm for CMRE is summarized in 1. We note that for our experiments, we split $\mathcal{D}$ such that the data used to train $f_a(c, x)$ in equation 4 is different than the data used to estimate the MR in equation 6. Specifically, 80% of the data in $\mathcal{D}$ is used to train $f_a(c, x)$ in equation 4 and the other 20% is used to estimate $\hat{\tau}_a'$, $\hat{\tau}_a$, and $\hat{\delta}_a'$.

## E.2   NMRE

NMRE adopts a similar strategy to CMRE for estimating the misreporting rate. Specifically, it leverages the differences in causal effect estimates. However, the key distinction between NMRE and CMRE is that NMRE doesn't account for potential confounders or treatment effect modifiers between $X^*$ and $Y$. As a result, NMRE employs a simple difference-in-means estimator to estimate

---

**Algorithm 1** CMRE algorithm

---

**Input:** $\mathcal{D} = \{(x_i, y_i, c_i, a_i)\}_i^N$ and $\mathcal{D}^* = \{(x_i^*, y_i, c_i)\}_i^M$
**Output:** $\widehat{MR}$, an estimate of the MR for each agent
  **for** each agent $a$ **do**
     Estimate $\theta_a(c)$ using equation 4
     Estimate $\theta^*(c)$ using equation 5
     Estimate $\hat{\tau}_a'$, $\hat{\tau}_a$, and $\hat{\delta}_a'$ using equation 6
     **return** $\frac{\hat{\tau}_a' - \hat{\tau}_a}{\hat{\delta}_a'}$
  **end for**

---

the average effect of the feature on $Y$ over both $\mathcal{D}^*$ and $\mathcal{D}$. Therefore, to estimate the MR for a given agent $a$, we define

$$\hat{\tau}' = \frac{1}{M_1} \sum_{i: i \in \mathcal{D}^*, x_i^* = 1} y_i - \frac{1}{M_0} \sum_{i: i \in \mathcal{D}^*, x_i^* = 0} y_i \tag{7}$$

and

$$\hat{\tau}_a = \frac{1}{N_{a1}} \sum_{\substack{i: i \in \mathcal{D}, x_i = 1, \\ a_i = a}} y_i - \frac{1}{N_{a0}} \sum_{\substack{i: i \in \mathcal{D}, x_i = 0, \\ a_i = a}} y_i, \tag{8}$$

where $M_x$ is the number of datapoints in $\mathcal{D}^*$ where $X^* = x$.

The misreporting rate is the estimated as:

$$\hat{MR} = \frac{\hat{\tau}' - \hat{\tau}_a}{\hat{\tau}'}.$$

The complete algorithm for NMRE is summarized in Algorithm 2.

---

**Algorithm 2** NMRE algorithm

---

**Input:** $\mathcal{D} = \{(x_i, y_i, c_i, a_i)\}_i^N$ and $\mathcal{D}^* = \{(x_i^*, y_i, c_i)\}_i^M$
**Output:** $\widehat{MR}$, an estimate of the MR for each agent
  **for** each agent $a$ **do**
     Estimate $\hat{\tau}'$ using equation 7
     Estimate $\hat{\tau}_a$ using equation 8
     **return** $\frac{\hat{\tau}' - \hat{\tau}_a}{\hat{\tau}'}$
  **end for**

---

### E.3 NDEE

The NDEE baseline estimates the misreporting rate by computing a quanitity similar to the natural direct effect of $A$ on $X$, divided by the probability $P_a(X = 1)$. Specifically, we estimate

$$\frac{1}{P_a(X = 1)} \int_C (\mathbb{E}_{P_a}[X|C = c] - \mathbb{E}_{P^*}[X|C = c]) P_a(C = c) dC. \tag{9}$$

Assuming that all datapoints in $P^*$ are generated a single trustworthy agent $a^*$, the integral term,

$$\int_C (\mathbb{E}_{P_a}[X|C = c] - \mathbb{E}_{P^*}[X|C = c]) P_a(C = c) dC,$$

can be interpreted as the natural direct effect of $A$ of $X$ within the treated population (i.e., data points where $A = a$ are the treated whereas $a^*$ refers to the untreated), when $C$ controls for all mediators and confounders between $A$ and $X$. We next show how to estimate the NDEE in practice, which is as follows.

**Dataset Preparation.** We modify the original dataset $\mathcal{D}^*$ such that $\mathcal{D}^* = \{(x_i^*, y_i, c_i, a_i)\}_{i=1}^M$ where $a_i = a^*$ for all $a_i$. Next, we combine both $\mathcal{D}$ and $\mathcal{D}^*$ to create a unified dataset:

$$\mathcal{D}' = \mathcal{D} \cup \mathcal{D}^*.$$

**Causal Effect Estimation** Let $\mathcal{F}$ denote a suitable function class and $\ell$ a loss function. We then learn a function

$$f(c, a) := \arg \min_{f' \in \mathcal{F}} \frac{1}{N + M} \sum_{i : i \in \mathcal{D}'} \ell(f'(c_i, a_i), x_i). \tag{10}$$

Next, we estimate the natural direct effect of $A$ over the treated population as follows:

$$\hat{\tau}_{\text{NDE}} := \frac{1}{N_a} \sum_{i : i \in \mathcal{D}, a_i = a} f(c_i, a) - f(c_i, a^*). \tag{11}$$

**Probability Estimation** The probability $P_a(X = 1)$ can be estimated simply as

$$\pi_a := \frac{1}{N_a} \sum_{i : i \in \mathcal{D}, a_i = a} x_i \tag{12}$$

The full algorithm is summarized in 3. The model $f(c, a)$ is implemented using XGBoost, where we use the default hyperparameters provided by the XGBoost library in Python [7] (a learning rate of 0.3, a maximume tree depth of 6, and L2 regularization with a coefficient of 1).

---

**Algorithm 3** NDEE algorithm
---
**Input:** $\mathcal{D} = \{(x_i, y_i, c_i, a_i)\}_i^N$, $\mathcal{D}^* = \{(x_i^*, y_i, c_i)\}_i^M$, and $\mathcal{D}' = \{(x_i, y_i, c_i, a_i)\}_i^{N+M}$
**Output:** $\widehat{\text{MR}}$, an estimate of the MR for each agent
   **for** each agent $a$ **do**
      Estimate $f(c, a)$ using equation 10
      Estimate $\hat{\tau}_{\text{NDE}}$ using equation 11
      Estimate $\pi_a$ using equation 12
      **return** $\frac{1}{\pi_a} \hat{\tau}_{\text{NDE}}$
   **end for**

---

**When can the NDEE accurately estimate the MR?** We show that if $A$ does not directly causally effect $X^*$, it is possible to obtain an accurate estimate of the misreporting rate using the NDEE. To show this, we can rewrite the misreporting rate as follows:

$$
\begin{aligned}
\text{MR} &= P_a(X^* = 0 | X = 1) \\
&= \frac{P_a(X = 1) - P_a(X^* = 1)}{P_a(X = 1)} \\
&= \frac{1}{P_a(X = 1)} (\mathbb{E}_{P_a}[X] - \mathbb{E}_{P_a}[X^*]) \\
&= \frac{1}{P_a(X = 1)} \int_C (\mathbb{E}_{P_a}[X | C = c] - \mathbb{E}_{P_a}[X^* | C = c]) P_a(C = c) dC.
\end{aligned}
$$

Thus, unless $\mathbb{E}_{P_a}[X^* | C = c] = \mathbb{E}_{P^*}[X^* | C = c]$, the NDEE will give a biased estimate of the misreporting rate. This equality will hold if $X^* \perp A | C$, which can only be true if $A$ does not directly affect $X^*$. Therefore, we should expect the NDEE to only work in settings where agents do not directly genuinely adapt their features.

### E.4 OC-SVM

Under our assumptions, no data points where $X = 0$ are misreported. Therefore, we restrict the OC-SVM approach to a subset of the data from $\mathcal{D}^*$ and $\mathcal{D}$ where $X = 1$, which we denote as $\mathcal{D}_1^*$ and $\mathcal{D}_1$, respectively. To train a One-Class SVM model, we use data from $\mathcal{D}_1^*$, which is assumed to contain no misreported instances. The One-Class SVM model, denoted as $g(y, c)$, is trained to identify outliers/misreported instances using only the variables $Y$ and $C$. The model outputs a 1 if a datapoint is classified an outlier, and 0 otherwise.

For a given agent $a$, we estimate the misreporting rate using the OC-SVM as:

$$\hat{\text{MR}} = \frac{1}{N_{a1}} \sum_{i:i \in \mathcal{D}_1, a_i = a} g(y_i, c_i).$$

We use the One-Class SVM implementation from the scikit-learn library [35]. Given our assumption that all data points in $\mathcal{D}_1^*$ are correctly reported, we used a small $\nu$ parameter (0.01). Additionally, we use an RBF kernel with a bandwidth parameter $\gamma = 0.1$.

## F  Additional Experiments

### F.1  Medicare Experiments

Figure 5 presents our Medicare experiments including the results from the OC-SVM estimator. The estimated misreporting rate for the OC-SVM is consistent across all HCC codes and agents, reflecting the results from our semi-synthetic loan dataset experiments. This suggests that the OC-SVM is unable to distinguish misreported data points from normal data points.

Tables 1 and 2 provide information about each of the nonpayment and payment HCCs that had $\hat{\delta}'_a > 0$ and were present in at least 1% of the switcher enrollees. For each HCC code, the tables report the estimated MR using CMRE, the number of stayer and switcher enrollees in year $t$, $\hat{\delta}'_a$, and the lower and upper bounds for the 95% confidence interval. We exclude HCCs that were nonpayment in year $t$ but were used as payment HCCs for the risk adjustment model in year $t + 1$, due to the potential incentive for agents to misreport them.

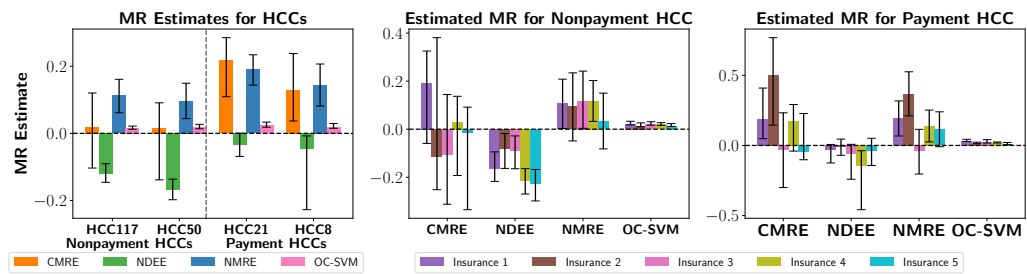

Figure 5: For each plot, the $y$-axis represents the estimated MR for an HCC code and the error bars represent a 95% confidence interval. **(Left)** The $x$-axis has two nonpayment HCCs (HCC117 and HCC50) and two payment HCCs (HCC21 and HCC8). Our approach (CMRE) has a MR estimate close to zero for nonpayment HCCs and significantly above zero for the payment HCCs, which aligns with what is expected in current literature. Baselines that fail to distinguish genuine adaptation from strategic adaptation (NDEE) seem to underestimate the MR and baselines that do not control for confounding (NMRE) seem to overestimate the MR. OC-SVM has a similar estimate for each HCC, making it ineffective at identifying misreported data points. **(Middle and Right)** The $x$-axis represents the baselines. The middle plot represents estimates for HCC50 and the right plot represents MR estimates for HCC8 across different private insurers (agents). Similar to the left plot, NDEE seems to underestimate the MR across most agents, and NMRE overestimates, and the MR estimates for OC-SVM are consistent across all agents and HCC codes.

Table 1: Nonpayment HCCs with $\hat{\delta}'_a > 0.1$ and present in at least 1% of switcher enrollees.

| HCC | Full Name | MR | # Stayers | # Switchers | $\hat{\delta}'_a$ | LCB | UCB |
|---|---|---|---|---|---|---|---|
| 50 | Delirium and Encephalopathy | .015 | 32294 | 4535 | .175 | -.138 | .091 |
| 117 | Pleural Effusion/Pneumothorax | .019 | 39037 | 5601 | .153 | -.103 | .120 |

Table 2: Payment HCCs with $\hat{\delta}'_a > 0.1$ and present in at least 1% of switcher enrollees.

| HCC | Full Name | MR | # Stayers | # Switchers | $\hat{\delta}$ | LCB | UCB |
|---|---|---|---|---|---|---|---|
| 8 | Metastatic Cancer and Acute Leukemia | .130 | 18762 | 2646 | .276 | .037 | .238 |
| 21 | Protein-Calorie Malnutrition | .217 | 21460 | 3338 | .270 | .109 | .285 |
| 84 | Cardio-Respiratory Failure and Shock | .046 | 40308 | 6292 | .247 | -.030 | .103 |
| 188 | Artificial Openings for Feeding or Elimination | .004 | 10528 | 1684 | .194 | -.110 | .210 |
| 2 | Septicemia, Sepsis, SIRS, and Shock | .112 | 28297 | 4309 | .190 | .040 | .248 |
| 135 | Acute Renal Failure | -.004 | 49021 | 7868 | .130 | -.073 | .116 |
| 103 | Hemiplegia/Hemiparesis | .241 | 12589 | 2395 | .129 | .066 | .439 |
| 86 | Acute Myocardial Infarction | .033 | 20555 | 3230 | .117 | -.121 | .217 |

## F.2 Loan Dataset Experiments

We conduct additional experiments using alternative versions of the loan fraud dataset to show how well our method and the baselines work under the DAGs defined in Figure 4. We also include two additional baselines that were not in the main paper: NDEE (no C) and NDEE (no S). Unlike the standard NDEE model, which controls for all covariates, NDEE (no C) doesn't control for confounders between $X^*$ and $Y$, and NDEE (no S) doesn't control for common causes of $A$ and $X^*$, e.g., $S$. These variants are used to highlight the importance of controlling for $S$ for NDEE.

### F.2.1 Simulation 1

The first simulation replicates the setup used to generate the results in Section 5. It includes four confounders of $X^*$ and $Y$: education ($C_E$), sex ($C_S$), marriage ($C_M$), and age ($C_A$). Among these variables, sex and marriage also causally effect $A$, reflecting a similar scenario represented by the DAG in Figure 4(g). The simulation details are provided below:

$$A_i \sim \text{Bernoulli}(0.05 + 0.3(1 - C_{Si}) + 0.3(1 - C_{Mi})),$$
$$X_i^* \sim \text{Bernoulli}(0.05 + 0.05 C_{Ei} + 0.3 C_{Si} C_{Mi} + 0.1 C_{Ai}^2 + \beta_A A_i),$$
$$Y_i \sim \text{Bernoulli}(0.05 + 0.05 C_{Ei} + 0.3 C_{Si} C_{Mi} + 0.1 C_{Ai}^2 + \beta_{X^*} X_i^*),$$
$$X_i \sim X_i^* + A_i(1 - X_i^*)\text{Bernoulli}(\mu),$$

In this simulation, NDEE (no S) doesn't control for either $C_S$ or $C_M$. Our main method, CMRE, controls for all covariates as they are all confounders between $X^*$ and $Y$. $\beta_A = 0.3$ and $\beta_{X^*} = 0.4$ unless specified otherwise. The results for this simulation are shown in Figure 6.

### F.2.2 Simulation 2

The second simulation includes three confounders of $X^*$ and $Y$: education ($C_E$), sex ($C_S$), and age ($C_A$). Marriage ($C_M$) is a common cause of $A$ and $X^*$ and an agent genuinely adapts eduction, reflecting similar scenarios represented by the DAGs in Figure 4(a) and 4(b). In addition, eduction is

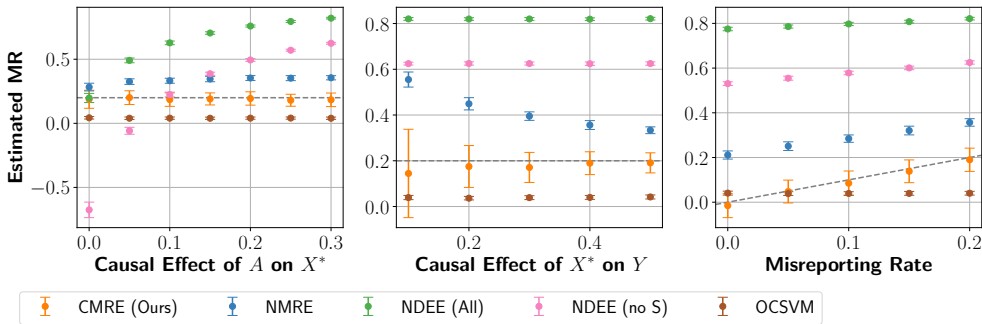

Figure 6: The $x$-axis is the causal effect of $A$ on $X^*$ (**left**), causal effect of $X^*$ on $Y$ (**middle**), and the misreporting rate (**right**). The $y$-axis is the estimated misreporting rate. Dashed lines represent the true misreporting rate and the error bars represent the standard deviation. Our approach (CMRE) accurately estimates the MR for all levels of genuine adaptation, the causal effect of $X^*$ on $Y$, and misreporting rates. The variance for our estimator depends on the magnitude of the causal effect of $X^*$ on $Y$. Baselines that do not adjust for confounding (NMRE) or do not distinguish between genuine adaptation and misreporting (NDEE) give biased estimates in various cases. NDEE is accurate when there is no genuine adaptation whereas NDEE (no S) is not, highlighting the need for controlling for common causes of $A$ and $X^*$. Anomaly detection methods (OC-SVM) are unable to distinguish misreported data points from unmanipulated data points.

also a treatment effect modifier. The simulation details are provided below:

$$A_i \sim \text{Bernoulli}(0.05 + 0.4(1 - C_{Mi})),$$
$$C'_{Ei} \sim C_{Ei} + (1 - C_{Ei})A_i\text{Bernoulli}(\beta_M),$$
$$X_i^* \sim \text{Bernoulli}(0.05 + 0.25C_{Mi} + 0.1C'_{Ei}C_{Si} + 0.1C_{Ai}^2 + \beta_A A_i),$$
$$Y_i \sim \text{Bernoulli}(0.05 + 0.2C'_{Ei}C_{Si} + 0.1C_{Ai}^2 + (\beta_{X^*} + 0.1C'_{Ei})X_i^*),$$
$$X_i \sim X_i^* + A_i(1 - X_i^*)\text{Bernoulli}(\mu),$$

In this simulation, NDEE (no S) doesn't control for $C_M$ and NDEE (no C) only controls for $C_M$. Our main method, CMRE, only controls for $C_E$, $C_S$, and $C_A$. $\beta_A = 0.1$, $\beta_M = 0.2$, and $\beta_{X^*} = 0.4$ unless specified otherwise. The results for this simulation are shown in Figure 7.

### F.2.3 Simulation 3

The third simulation includes three confounders of $X^*$ and $Y$: education ($C_E$), sex ($C_S$), and age ($C_A$). Marriage ($C_M$) is a common cause of $A$ and $Y$ and an agent genuinely adapts education, reflecting the scenario represented by the DAG in Figure 4(c). In addition, education is also a treatment effect modifier. $\beta_A = 0.1$, $\beta_M = 0.2$, and $\beta_{X^*} = 0.4$ unless specified otherwise. The simulation details are provided below:

$$A_i \sim \text{Bernoulli}(0.05 + 0.4(1 - C_{Mi})),$$
$$C'_{Ei} \sim C_{Ei} + (1 - C_{Ei})A_i\text{Bernoulli}(\beta_M),$$
$$X_i^* \sim \text{Bernoulli}(0.05 + 0.1C'_{Ei}C_{Si} + 0.1C_{Ai}^2 + \beta_A A_i),$$
$$Y_i \sim \text{Bernoulli}(0.05 + 0.2C_{Mi} + 0.1C'_{Ei}C_{Si} + 0.05C_{Ai}^2 + (\beta_{X^*} + 0.1C'_{Ei})X_i^*),$$
$$X_i \sim X_i^* + A_i(1 - X_i^*)\text{Bernoulli}(\mu),$$

In this simulation, NDEE (no C) only controls for $C_M$. Our main method, CMRE, only controls for $C_E$, $C_S$, and $C_A$. The results for this simulation are shown in Figure 8.

### F.2.4 Simulation 4

The fourth simulation includes two confounders of $X^*$ and $Y$: sex ($C_S$), and age ($C_A$). Marriage ($C_M$) is a common cause of $A$ and $Y$ and an agent genuinely adapts education, which is a mediator

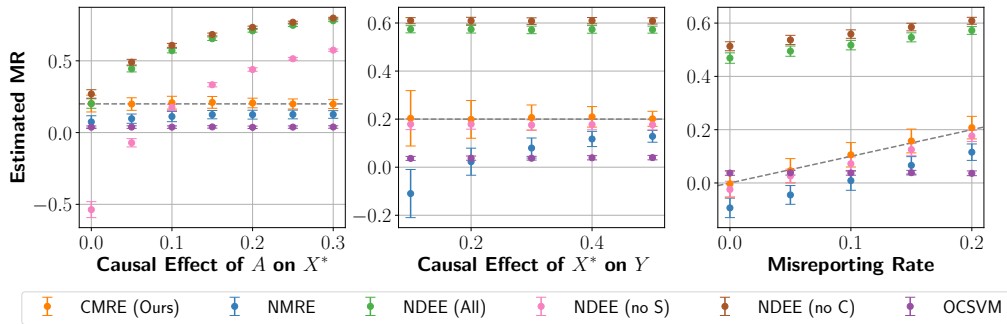

Figure 7: The $x$-axis is the direct causal effect of $A$ on $X^*$ **(left)**, causal effect of $X^*$ on $Y$ **(middle)**, and the misreporting rate **(right)**. The $y$-axis is the estimated misreporting rate. Dashed lines represent the true misreporting rate and the error bars represent the standard deviation. Our approach (CMRE) accurately estimates the MR for all levels of genuine adaptation, the causal effect of $X^*$ on $Y$, and misreporting rates. The variance for our estimator depends on the magnitude of the causal effect of $X^*$ on $Y$. Baselines that do not adjust for confounding (NMRE) or do not distinguish between genuine adaptation and misreporting (NDEE) give biased estimates in various cases. NDEE is accurate when there is no genuine adaptation whereas NDEE (no S) and NDEE (no C) are not, as they either don't control for common causes of $A$ and $X^*$ or mediators of $A$ and $X^*$. Anomaly detection methods (OC-SVM) are unable to distinguish misreported data points from unmanipulated data points.

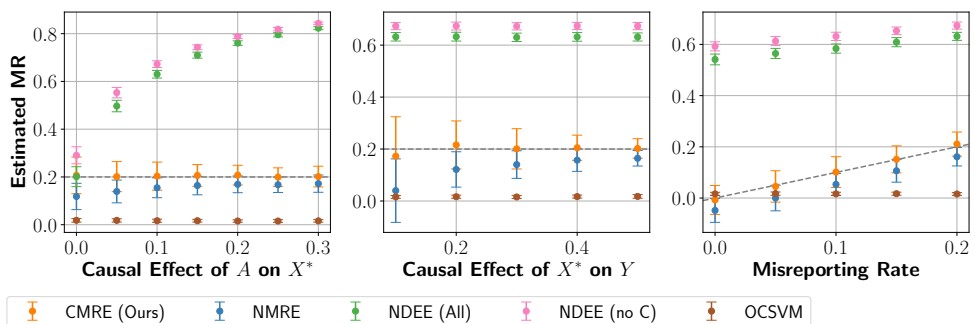

Figure 8: The $x$-axis is the direct causal effect of $A$ on $X^*$ **(left)**, causal effect of $X^*$ on $Y$ **(middle)**, and the misreporting rate **(right)**. The $y$-axis is the estimated misreporting rate. Dashed lines represent the true misreporting rate and the error bars represent the standard deviation. Our approach (CMRE) accurately estimates the MR for all levels of genuine adaptation, the causal effect of $X^*$ on $Y$, and misreporting rates. The variance for our estimator depends on the magnitude of the causal effect of $X^*$ on $Y$. Baselines that do not adjust for confounding (NMRE) or do not distinguish between genuine adaptation and misreporting (NDEE) give biased estimates in various cases. NDEE is accurate when there is no genuine adaptation whereas NDEE (no C) is not, as it doesn't account for the mediators of $A$ and $X^*$. Anomaly detection methods (OC-SVM) are unable to distinguish misreported data points from unmanipulated data points.

of $A$ and $X^*$, reflecting the scenario represented by the DAGs in Figure 4(d) and 4(e). The simulation details are provided below:

$$A_i \sim \text{Bernoulli}(0.05 + 0.4(1 - C_{Mi})),$$
$$C'_{Ei} \sim C_{Ei} + (1 - C_{Ei})A_i\text{Bernoulli}(\beta_M),$$
$$X_i^* \sim \text{Bernoulli}(0.05 + 0.3C'_{Ei}C_{Si} + 0.1C_{Ai}^2 + \beta_A A_i),$$
$$Y_i \sim \text{Bernoulli}(0.05 + 0.2C_{Mi} + 0.1C_{Si} + 0.05C_{Ai}^2 + \beta_{X^*}X_i^*),$$
$$X_i \sim X_i^* + A_i(1 - X_i^*)\text{Bernoulli}(\mu),$$

In this simulation, NDEE (no C) only controls for $C_M$ and $C_E$. Our main method, CMRE, only controls for $C_S$ and $C_A$. $\beta_A = 0.1$, $\beta_M = 0.2$, and $\beta_{X^*} = 0.4$ unless specified otherwise. The results for this simulation are shown in Figure 9.

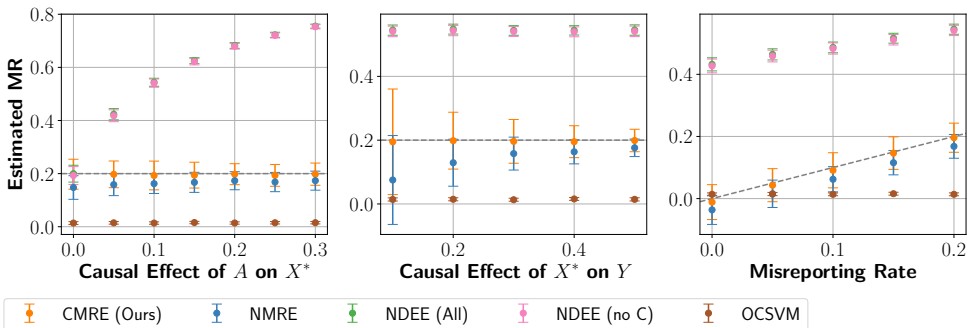

Figure 9: The $x$-axis is the direct causal effect of $A$ on $X^*$ (**left**), causal effect of $X^*$ on $Y$ (**middle**), and the misreporting rate (**right**). The $y$-axis is the estimated misreporting rate. Dashed lines represent the true misreporting rate and the error bars represent the standard deviation. Our approach (CMRE) accurately estimates the MR for all levels of genuine adaptation, the causal effect of $X^*$ on $Y$, and misreporting rates. The variance for our estimator depends on the magnitude of the causal effect of $X^*$ on $Y$. Baselines that do not adjust for confounding (NMRE) or do not distinguish between genuine adaptation and misreporting (NDEE) give biased estimates in various cases. Both NDEE and NDEE (no C) are accurate when there is no genuine adaptation, as they control for all common causes of $A$ and $X^*$ and mediators of $A$ and $X^*$. Anomaly detection methods (OC-SVM) are unable to distinguish misreported data points from unmanipulated data points.

### F.2.5   Simulation 5

The fifth simulation includes two confounders of $X^*$ and $Y$: sex ($C_S$), and age ($C_A$). Marriage ($C_M$) is a common cause of $A$ and $X^*$ and an agent genuinely modifies education, which is a mediator of $A$ and $X^*$, reflecting similar scenarios represented by the DAGs in Figure 4(d) and 4(f). The simulation details are provided below:

$$A_i \sim \text{Bernoulli}(0.05 + 0.4(1 - C_{Mi})),$$
$$C'_{Ei} \sim C_{Ei} + (1 - C_{Ei})A_i\text{Bernoulli}(\beta_M),$$
$$X_i^* \sim \text{Bernoulli}(0.05 + 0.2C_{Mi} + 0.3C'_{Ei}C_{Si} + 0.1C_{Ai}^2 + \beta_A A_i),$$
$$Y_i \sim \text{Bernoulli}(0.05 + 0.3C_{Si} + 0.05C_{Ai}^2 + \beta_{X^*}X_i^*),$$
$$X_i \sim X_i^* + A_i(1 - X_i^*)\text{Bernoulli}(\mu),$$

In this simulation, NDEE (no C) only controls for $C_M$ and $C_E$ and NDEE (no S) doesn't control for $C_M$. Our main method, CMRE, only controls for $C_S$ and $C_A$. $\beta_A = 0.1$, $\beta_M = 0.2$, and $\beta_{X^*} = 0.4$ unless specified otherwise. The results for this simulation are shown in Figure 10.

## G   Software and Hardware

### G.1   Software

All of the code for the experiments was written in Python 3.10.16 (PSF License). The XGBoost models were implemented using the XGBoost 2.1.4 (Apache License 2.0) [7]. The OC-SVM baseline was implemented by using scikit-learn 1.6.1 (BSD License) [35], which used the implementation of the One-Class SVM. To generate the semi-synthetic datasets and for data processing tasks, both numpy 2.0.2 (modified BSD license) [16] and pandas 2.2.3 (BSD license) [34] were employed. For the Medicare dataset, HCCPy 0.1.9 (Apache License 2.0) was employed to extract the HCCs from raw data. All plots were created using matplotlib 3.10.1 (PSF License) [20].

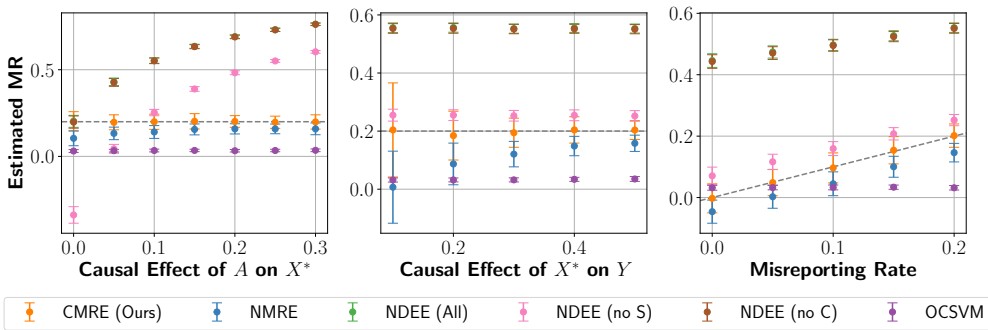

Figure 10: The $x$-axis is the direct causal effect of $A$ on $X^*$ **(left)**, causal effect of $X^*$ on $Y$ **(middle)**, and the misreporting rate **(right)**. The $y$-axis is the estimated misreporting rate. Dashed lines represent the true misreporting rate and the error bars represent the standard deviation. Our approach (CMRE) accurately estimates the MR for all levels of genuine adaptation, the causal effect of $X^*$ on $Y$, and misreporting rates. The variance for our estimator depends on the magnitude of the causal effect of $X^*$ on $Y$. Baselines that do not adjust for confounding (NMRE) or do not distinguish between genuine adaptation and misreporting (NDEE) give biased estimates in various cases. Both NDEE and NDEE (no C) are accurate when there is no genuine adaptation, as they control for all common causes of $A$ and $X^*$ and mediators of $A$ and $X^*$. In contrast, NDEE does not control for common causes of $A$ and $X^*$, which makes it biased. Anomaly detection methods (OC-SVM) are unable to distinguish misreported data points from unmanipulated data points.

## G.2 Hardware

All experiments were conducted using 16 CPU cores and 32 GB of memory on a computing cluster with 2 x 2.5 GHz Intel Haswell (Xeon E5-2680v3) processors, which was managed using a Slurm resource manager. The simulations for all of the five semi-synthetic loan experiments took approximately 5 hours to complete, whereas the experiments over the Medicare dataset took approximately 36 hours to complete.

