# OpenReview forum: "Disentangling misreporting from genuine adaptation in strategic settings: a causal approach"
_NeurIPS.cc/2025/Conference — NeurIPS 2025 poster_

### Official Review · Reviewer_pJAD · 2025-07-02

**Clarity:** 3
**Significance:** 3
**Originality:** 3
**Rating:** 4
**Confidence:** 3

**Summary:**

This paper addresses the challenge of distinguishing between genuine modification and misreporting of features by agents in settings where machine learning models are used for resource allocation, such as in loan applications or government insurance payouts. Agents may either genuinely change their behavior (genuine modification) or simply report false features (misreporting) to obtain better outcomes. The authors propose a causally-motivated approach to identify and quantify the average rate of misreporting by leveraging the fact that misreported features, unlike genuinely modified ones, do not causally affect downstream variables. By comparing the causal effect of features on their descendants in both manipulated (potentially misreported) and unmanipulated datasets, they develop an estimator for the misreporting rate. The paper provides theoretical guarantees for the identifiability and variance of their estimator, and demonstrates its effectiveness with experiments on both semi-synthetic and real Medicare datasets.

**Questions:**

See Weakness.

**Ethical Concerns:**

["NO or VERY MINOR ethics concerns only"]

**Final Justification:**

I sincerely appreciate the authors’ effort in providing such a detailed and comprehensive response. I will keep the positive rating.

**Quality:**

3

**Strengths And Weaknesses:**

1. Novel Causal Framework: The paper introduces a new causal approach for disentangling genuine modification from misreporting, providing a theoretically grounded solution to a longstanding problem in strategic behavior and resource allocation settings.

2. Theoretical Guarantees: The authors rigorously prove the identifiability of the misreporting rate and analyze the variance of their estimator, offering strong theoretical justification and transparency for their method.

3. Empirical Validation: The approach is thoroughly validated on both semi-synthtic and real-world datasets, demonstrating its practical utility and superior performance.


*Weakness*

1. Strong Assumptions Needed: The method relies on some pretty strong assumptions, like no unobserved confounding, which might not always hold in real-world data. If those assumptions are violated, the results could be off.

2. Requires Special Datasets: You need access to both manipulated (possibly misreported) and unmanipulated datasets. In practice, getting a “clean” dataset where no one has any incentive to misreport isn’t always realistic.

3. Limited Scope for Complex Behavior: The framework mostly handles binary features and assumes misreporting only happens in one direction. It doesn’t really cover more complex or nuanced types of misreporting, so its scope is a bit limited.

---

> ### Author Rebuttal · Authors · 2025-07-30
>
> We thank the reviewer for their insightful feedback. We are encouraged that the reviewer found our causal framework novel and well-motivated, our theoretical analysis rigorous, and our empirical experiments compelling.
>
> **1 - No unobserved confounding assumption.** The reviewer is correct in that the no unobserved confounding assumption is untestable. However, it is standard in causal inference literature and even causally-motivated work on strategic adaptation [1]. In addition, our method is compatible with existing sensitivity analysis techniques (e.g., [2]) that can relax Assumption 3 and provide bounds on the misreporting rate. Specifically, one can apply sensitivity analysis to get bounds on the estimated causal effects $\delta'_a$, $\tau'_a$, and $\tau_a$, and then apply those bounds through Theorem 1 to obtain a bound on the misreporting rate. We will highlight sensitivity to confounding as a promising direction for future work in the final version.
>
> **2 - Access to a manipulated dataset might be unrealistic.** The reviewer is correct in that our approach requires access to an unmanipulated dataset. We note that such data is often available from pre-deployment (where there may be no incentive to misreport) or from audits.
>
> **3 - The framework mostly handles binary features and assumes misreporting only happens in one direction.**
>
>
> We note our method assumes only that $X^\ast$ and $X$ are binary; other variables ($C, A$, $Y$) can be arbitrarily defined. We chose this setup since it allows for a straightforward definition of the misreporting rate. However, our approach could potentially be extended to multi-valued or continuous features with additional parametric and behavioral assumptions (e.g., partially linear models and agents misreporting $X^\ast$ by applying a fixed additive bias, such that $X = X^\ast + \delta$ for some constant $\delta$). We will add this as a promising direction for future work in the final version.
>
> We also note that while we assume agents only misreport in one direction (e.g., when $X^\ast=0$), our framework can be easily adapted to the opposite case (e.g., misreporting only when $X^\ast=1$). Moreover, one-directional misreporting is a realistic assumption -- e.g., in the Medicare context, private insurers are always incentivised to overreport diagnoses to increase reimbursement and have no incentive to underreport. Similar incentive asymmetries exist in school admissions, loan applications, and health insurance quotes (e.g., GPA, employment, health).
>
>
> **Citations**
>
> [1] Chang, Trenton, et al. "Who’s gaming the system? a causally-motivated approach for detecting strategic adaptation." Advances in Neural Information Processing Systems, 2024
>
> [2] Kallus, Nathan, Xiaojie Mao, and Angela Zhou. "Interval estimation of individual-level causal effects under unobserved confounding." The 22nd international conference on artificial intelligence and statistics. PMLR, 2019.

---

> > ### Comment · Reviewer_pJAD · 2025-08-07
> >
> > Thanks for your response. I will maintain my score.

---

### Official Review · Reviewer_YpE8 · 2025-07-02

**Clarity:** 3
**Significance:** 3
**Originality:** 3
**Rating:** 4
**Confidence:** 4

**Summary:**

This work proposes a causal identification and estimation framework for characterizing settings in which agents may strategically manipulate their characteristics to obtain improved outputs. In particular, the authors disentangle to forms of agent responses: genuine modification, in which an agent alters characteristics in such a way that changes their downstream outcomes, and misreporting, in which agents report incorrect features without a change in their underlying features. The paper establishes the identifying conditions under which the misreporting rate can be identified via a corpus with no misreporting and genuine modification + misreporting. The authors propose an estimator for this estimated and analyze its variance properties, then validate it via synthetic and semi-synthetic experiments.

**Questions:**

In general I think this work makes a valuable contribution, but have concerns in light of my review above. I will raise my score if the questions below are addressed in a satisfactory manner.

- How robust is the theoretical and experimental results to the the misreporting rate is assumed to only depend on the true feature values $X^{*}$? How easily can the framework be extended to support subgroup-dependent misreporting rates?

- To what extent are values of the $\beta$ range reported in Figure 2 reasonable? More concretely, what is the estimated $\beta$ value in the experiments with real data, as a reference point?


- How robust are the real-data experimental findings to the exclusion conditions designed to reduce variance, and why are these exclusion conditions reasonable? How robust are the synthetic results to coefficient values?

**Ethical Concerns:**

["NO or VERY MINOR ethics concerns only"]

**Final Justification:**

The authors have addressed my concerns regarding the scope of the framework and the experiment design. I remain concerned about the variance of the estimator. I raise my score accordingly.

**Limitations:**

Yes.

**Paper Formatting Concerns:**

None.

**Quality:**

2

**Strengths And Weaknesses:**

## Significance.

The problem of strategic response to incentives is timely and important. The authors clearly articulate the relevance of this problem via a well-motivated example involving the U.S. Medicare Advantage (MA) program, and show that the proposed approach can be used to address policy-relevant estimation problems in this setting. This is a strong example of use case inspired basic research, that likely generalizes to other similar settings.

## Originality.

The authors clearly situate the proposed work in the context of other related literature in this space and articulate the technical novelty.

Minor: line 74 claims that prior work by Chang et al. [5] can only partially identify misreporting rates - this is not a limitation of their approach, per se, if this existing framework depends on weaker assumptions. Please clarify this in the next version of the draft.


## Clarity.

The work is generally clear and easy to follow. The use of a running example throughout the work is effective in motivating the technical components.

Minor:
- The use of observed and true features in potential outcome notation $X(X^*=x^*)$ (line 119) is slightly awkward. I do not see this frequently used throughout the text. Consider revising this notation if possible.
- In Section 4.1, the build-up to Theorem 1 is somewhat long. Consider shifting Theorem 1 towards the beginning of the section to communicate the key idea then draw connections to Lemma 1 after for intuition. If I am following correctly, Thm 1 does not inherit a direct dependence on Lemma 1 and Lemma 1 is included for pedagogical purposes.
- Line 206: extra equation number.
- Lines 144-152. Consider translating this text description of the limitations of status quo estimation approaches into a compelling main figure to clearly convey why the proposed approach is important over a naive baseline. This figure might include an illustrative empirical result from the appendix.

## Quality.

This work generally meets standards for a causal inference paper with the stated objectives. Assumptions are, for the most part, clearly stated. The identification analysis is clean and interesting. The estimation approach and corresponding experiments are generally well-motivated. That said, I do have several key concerns that affect my scores:

### Restrictive modeling setup.

A key concern I have with the current setup is that the misreporting rate is assumed to only depend on the true feature values $X^\star$. This assumes that no features confound the relationship between $X^\star$ and $X$ - which is generally quite strong given the motivating example provided by the authors (e.g., misreporting could be higher for patients with certain medical conditions or coverage). Complicating maters further, this assumption seems to be a core technical requirement of the framework. Eg., the authors assume the misreported group will be a random sample of the group where $X^\star=0$ (line 172).

At minimum, this should be clearly acknowledged and explained in the preliminaries and limitations. Ideally, there should be further theoretical/experimental analysis probing robustness to this assumptions' violations. For instance, a few guiding questions are: does the existing analysis carry through if we condition on a confounding variable  $X<-V->X^*$ (where $V$ is the confounder) via a stratification approach? If we took such an approach, how would this affect variance of the estimator in cases where some strata have small treatment effects? Is there a way to test whether this independence assumption is violated in practice?

The assumption that $X$ and $X^\star$ are dichotomous is also restrictive, albeit less concerning than the assumption above.

On the other hand, the generality of the framework across settings Fig 1 (a-c) is a clear strength.

### Variance of the estimator.

As noted in Section 4.2, the proposed estimator will have high variance in cases where the true treatment effect in the unmanipulated dataset is small. We observe this empirically in Figure 2 (center), where confidence intervals are wide when the effect of $X^\star$ on Y is small. While some variance in the estimator is tolerable, I have concerns that the extent of the variance in these results may affect the substantive conclusions that can be drawn from the estimation approach. For instance, when $\beta_{X^\star}=0.1$, the interval spans roughly 0.03-0.4, which is quite large. To what extent are values of the $\beta_{X^\star}$ range reported in Figure 2 reasonable? More concretely, what is the estimated $\beta_{X^\star}$ value in the experiments with real data, as a reference point?

Further, we also observe wide confidence intervals with real data experiments. While the authors note that their approach is the only one that provides MR rates not statistically significantly different from zero (line 342), this is trivially satisfied by wide confidence intervals. For instance, the upper and lower bounds exceed those provided by NDEE and NMRE for Fig. 3 (center).

I have further concerns that the experimental setup may mask further variance issues that would arise under realistic application conditions. In particular, the authors report exclusion conditions (line 318-320) designed to reduce variance in estimates. What percent of records were omitted on the basis of these criteria? How do the results look with these filters omitted?

### Estimators & Experiment Setup.

The below are more minor points.

First, the authors leverage S-learners for estimation. While this is a reasonable starting point, this approach is likely to have larger variance than other DR approaches that apply a first-order bias correction. Expanding to DR estimators may improve the variance properties of the proposed estimator.

Second, the synthetic experiment with hard-coded coefficients may affect the results. Because this is (understandably) the only experiment with access to true misreporting rates, the setup here is important. The authors should report robustness experiments examining how results change under different D.G.P setups (or with randomly sampled coefficients).

---

> ### Author Rebuttal · Authors · 2025-07-30
>
> We thank the reviewer for their thoughtful and constructive feedback! The reviewer's comment that our work ``is a strong example of use case inspired basic research'' is truly encouraging. We are also encouraged that they found the problem timely and important, our technical novelty well articulated, and the work clear and easy to follow.
>
> **1 - How robust is the [work to the assumption that] the misreporting rate depends only on the true feature values $X^\ast$? How easily can the framework be extended to support subgroup-dependent misreporting rates?** We appreciate the reviewer's question and clarify two points: (1) our method already accommodates other variables that affect the misreporting rate, (2) it can be trivially extended to even more fine-grained subgroup-dependent misreporting rates.
>
> First, the agent variable $A$ can act as a confounder/effect modifier with $X \leftarrow A \rightarrow X^\ast$ exactly as the reviewer suggests (see figure 1a, b, and c). Our method is able to account for $A$, as reflected in the experiments section, figure 3 (middle and right), where we report misreporting rates conditional on $A=a$ (the insurance company). We acknowledge that lines 172-173 were unclear. Originally, these lines said "... only the variable $X^\ast$ influences an agent's decision to misreport..." A more precise statement is "Conditional on the agent identity, only $X^{*}$ influences the decision to misreport." That is, *conditional* on the agent, the misreported group is a random sample of those with $X^\ast=0$. We will clarify in the final version.
>
> Second, our method can be trivially extended to accommodate more granular subgroup analysis by estimating CATE versions of TAFR, NAFR, and TAFM estimands conditioned on any features of interest. While this might increase the variance, as the reviewer noted, it reflects a natural trade-off between granularity and statistical stability that can be navigated based on the application. We will include this discussion in the final version.
>
>
> **2 - Variance of the estimator** The reviewer is correct that our approach gives high variance estimates when the causal effect of $X^\ast$ on $Y$ is low. However, we view this as a strength -- it appropriately reflects uncertainty rather than projecting false confidence, which is critical in high-stakes settings.
>
> We also clarify that the non-significant estimated MR rates in Fig. 3 (center) are not simply due to wide intervals. As shown in Fig. 3 (left and right), our method detects significant misreporting when present, indicating that the intervals are informative. The wider intervals in real data reflect honest uncertainty, while NDEE and NMRE yield narrower but biased estimates. Thus, our intervals, though broader, lead to more reliable conclusions.
>
> We note our analysis gives principled approaches for reducing this variance: by picking downstream variables known to be significantly affected by the potentially misreported variable. For example, in our real data analysis, we used mortality as the downstream outcome for simplicity. However, many diagnoses have weak causal effects on mortality, leading to high variance. Using condition-specific downstream outcomes (e.g., insulin for diabetes-related diagnoses) would likely reduce variance. Further reductions in variance are possible, as the reviewer points out, by using DR estimators rather than S-learners. We used S-learners for simplicity. We will clarify these points in the final version.
>
> **3 - Are the $\beta$'s chosen for the simulation a good reference point for the real data?** We appreciate the reviewer’s question. The $\beta$ values were chosen to reflect effect sizes observed in the Medicare dataset. Our simulations used $\beta \in [0.1, 0.5]$, consistent with estimated effects shown in Appendix F.1, where each diagnosis’s effect on mortality ($Y$) falls between 0.117 and 0.276.
>
>
> **4 - How robust are the real-data findings to the exclusion [criteria], and why are [they] reasonable? What percent of the records were omitted on the basis of these criteria?** We thank the reviewer for the thoughtful question. No patient records were excluded; the criteria only filtered which of the 182 diagnoses were reported in the appendix table, not which patient records were used in the analysis. In other words, the reported estimates and their variances do not change at all when removing these exclusion criteria. We would just have 182 diagnoses reported rather than the top 10. These thresholds were selected with clinical input to ensure stable estimates when using mortality as the outcome by excluding diagnoses clinically known to have minimal impact on mortality (e.g., cataracts or osteoarthritis of the hip or knee). In our final version, we will consider reporting all diagnoses in the appendix.
>
>
> **5 - How robust are the synthetic results to coefficient values?** We chose coefficients so that the Bernoulli input remained in $[0, 1]$ without requiring transformations (e.g., sigmoid), allowing direct interpretability. Specifically, this allowed us to set the $X^\ast$ coefficient equal to the causal effect. As the reviewer suggests, an alternative is to draw coefficients randomly and apply a sigmoid to the input of the Bernoulli draw, offering more flexibility but requiring additional steps to recover causal effects. We conducted this analysis and found that our conclusions remain robust.
> The three tables below shows the results from the reviewer's suggested analysis; it shows the squared error of the estimated misreporting rate. Our approach outperforms the baselines in the majority of settings, and performs comparably (not statistically significantly different from) the best baseline in a few of the settings, as indicated by the bold entries in the tables.
>
>
>
> **6 - The assumption that $X, X^\ast$ are dichotomous is restrictive.** We chose this setup since it allows for a straightforward definition of the misreporting rate. However, our approach could potentially be extended to multi-valued or continuous features with additional parametric and behavioral assumptions (e.g., partially linear models and agents misreporting $X^\ast$ by applying a fixed additive bias, such that $X = X^\ast + \delta$ for some constant $\delta$). We will add this as a promising direction for future work in the final version.
>
>
>
> **7 -- Other minor comments**
> - **Prior work by Chang et al.**: We appreciate the reviewer's point and will clarify the differences with Chang et al.
> - **Notation on line 119 is slightly awkward**: We will revise the notation as suggested.
> - **Buildup to Theorem 1 is somewhat long**: We appreciate the reviewer's suggestions and will consider restructuring Section 4.1 to enhance the flow.
> - **Extra equation number**: We will remove.
> - **Convert text description to main figure**: We thank the reviewer for this suggestion! We agree that a visual comparison would clarify the limitations of naive estimators and will add a figure illustrating how our method separates misreporting from genuine modification using causal descendants.
>
>
> ### Table 1: MSE of Estimated MR vs Causal Effect of $A$ on $X^\ast$
>
> | Causal Effect of $A$ on $X^\ast$ | CMRE (Ours) | NMRE | NDEE | OCSVM |
> |---|---|---|---|---|
> | 0.000 | **0.003 ± 0.004** | 0.157 ± 0.022 | **0.000 ± 0.001** | 0.029 ± 0.002 |
> | 0.050 | **0.003 ± 0.004** | 0.126 ± 0.018 | 0.010 ± 0.003 | 0.028 ± 0.002 |
> | 0.100 | **0.003 ± 0.003** | 0.101 ± 0.016 | 0.031 ± 0.005 | 0.028 ± 0.003 |
> | 0.150 | **0.003 ± 0.003** | 0.084 ± 0.017 | 0.054 ± 0.007 | 0.028 ± 0.003 |
> | 0.200 | **0.003 ± 0.004** | 0.068 ± 0.018 | 0.077 ± 0.007 | 0.029 ± 0.003 |
> | 0.250 | **0.003 ± 0.005** | 0.060 ± 0.016 | 0.099 ± 0.007 | 0.029 ± 0.003 |
> | 0.300 | **0.003 ± 0.006** | 0.052 ± 0.018 | 0.118 ± 0.008 | 0.029 ± 0.003 |
>
> ### Table 2: MSE of Estimated MR vs Causal Effect of $X^\ast$ on $Y$
>
> | Causal Effect of $X^\ast$ on $Y$ | CMRE (Ours) | NMRE | NDEE | OCSVM |
> |---|---|---|---|---|
> | 0.100 | **0.034 ± 0.073** | 38.053 ± 220.579 | 0.118 ± 0.008 | **0.029 ± 0.003** |
> | 0.200 | **0.009 ± 0.019** | 0.060 ± 0.039 | 0.118 ± 0.008 | **0.029 ± 0.003** |
> | 0.300 | **0.004 ± 0.006** | 0.060 ± 0.024 | 0.118 ± 0.008 | 0.029 ± 0.002 |
> | 0.400 | **0.003 ± 0.006** | 0.052 ± 0.018 | 0.118 ± 0.008 | 0.029 ± 0.003 |
> | 0.500 | **0.002 ± 0.003** | 0.041 ± 0.012 | 0.118 ± 0.008 | 0.028 ± 0.003 |
>
> ### Table 3: MSE of Estimated MR vs Misreporting Rate
>
> | Misreporting Rate | CMRE (Ours) | NMRE | NDEE | OCSVM |
> |---|---|---|---|---|
> | 0.000 | **0.003 ± 0.003** | 0.073 ± 0.015 | 0.185 ± 0.012 | **0.001 ± 0.001** |
> | 0.050 | **0.003 ± 0.003** | 0.065 ± 0.015 | 0.167 ± 0.011 | **0.001 ± 0.000** |
> | 0.100 | **0.003 ± 0.003** | 0.058 ± 0.015 | 0.150 ± 0.010 | **0.005 ± 0.001** |
> | 0.150 | **0.003 ± 0.003** | 0.051 ± 0.014 | 0.134 ± 0.009 | 0.014 ± 0.002 |
> | 0.200 | **0.003 ± 0.006** | 0.052 ± 0.018 | 0.118 ± 0.008 | 0.029 ± 0.003 |

---

> > ### Comment · Reviewer_YpE8 · 2025-08-04
> >
> > **2 - Variance of the estimator**  Thank you, this partially addresses my concerns. In particular, I reman concerned that the variance of the estimator remains high for small Beta — eg. treatment effect reference range 0.117 and 0.276 mentioned by the authors. However, the authors raise a good point: baselines give biased estimates in this setting. In sum I remain concerned about variance and believe this may limit the practical utility of the approach but believe the work remains solid as a whole.
> >
> > Further, thank you for reporting these additional robustness checks.
> >
> > Finally, I remark that I am satisfied by the author's response to 9UrE's concern that the no unmeasured confounding assumption (Assumption 3) is untestable and potentially unrealistic in many real-world settings. Such assumptions are standard in similar papers and the sensitivity analysis offers a credible/standard workaround.
> >
> >  I raise my score accordingly.

---

### Official Review · Reviewer_9UrE · 2025-07-03

**Clarity:** 3
**Significance:** 3
**Originality:** 3
**Rating:** 4
**Confidence:** 3

**Summary:**

This paper addresses the problem of detecting and quantifying misreporting by strategic agents in settings where machine learning models are used for resource allocation decisions. The key challenge tackled is distinguishing between two types of strategic behavior: genuine modification and misreporting. The authors' core insight is that misreported features do not causally affect downstream variables, unlike genuinely modified features. Leveraging this asymmetry, they compare causal effects derived from manipulated datasets (where misreporting may occur) and unmanipulated datasets (where misreporting is absent) to quantify the misreporting rate.

**Questions:**

- How robust is the method to violations of the "no unobserved confounding" assumption (Assumption 3)?
- Can the method be generalized to continuous or categorical features?
- What are the barriers to deploying CMRE in practice (e.g., data access, model transparency)?

**Ethical Concerns:**

["NO or VERY MINOR ethics concerns only"]

**Final Justification:**

I have gone through other reviews and the corresponding responses. Most of my concerns have been addressed. This paper proposes a novel approach for estmating the extent to which strategic agents misreport their features. This setting is useful in practice such as credit. Overall, I recommend to accept this paper.

**Limitations:**

yes

**Quality:**

3

**Strengths And Weaknesses:**

Strengths
- The paper provides formal definitions, assumptions, and proofs for identifiability (Lemma 1, Theorem 1) with a clear mathematical treatment.
- Addresses a gap in the literature by formally distinguishing misreporting from genuine behavioral change.
- Using the asymmetric effect on causal descendants to identify misreporting is innovative.

Weaknesses
- The paper could better explain why misreporting doesn't affect causal descendants before diving into technical details.
- Identifies average misreporting rates but not individual ones.
- The no unmeasured confounding assumption (Assumption 3) is untestable and potentially unrealistic in many real-world settings.
- The reliance on causal descendants means the method may not apply to settings where downstream effects are unobservable or delayed.
- The approach is most useful in settings where agents have incentives to misreport and unmanipulated data is available, which may not cover all strategic ML scenarios.

---

> ### Author Rebuttal · Authors · 2025-07-30
>
> We thank the reviewer for their insightful feedback. We are encouraged that the reviewer values the rigor of our theoretical analysis, the gap our work addresses, and the innovation of our approach.
>
> **1 - No unobserved confounding assumption.** The reviewer is correct in that the no unobserved confounding assumption is untestable. However, it is standard in causal inference literature and even causally-motivated work on strategic adaptation [1]. In addition, our method is compatible with existing sensitivity analysis techniques (e.g., [2]) that can relax Assumption 3 and provide bounds on the misreporting rate. Specifically, one can apply sensitivity analysis to get bounds on the estimated causal effects $\delta'_a$, $\tau'_a$, and $\tau_a$, and then apply those bounds through Theorem 1 to obtain a bound on the misreporting rate. We will highlight sensitivity to confounding as a promising direction for future work in the final version.
>
>
>
> **2 - Can the method be generalized to continuous or categorical features?** We note our method assumes only that $X^\ast$ and $X$ are binary; other variables ($C$, $A$, $Y$) can be arbitrarily defined. We chose this setup since it allows for a straightforward definition of the misreporting rate. However, our approach could potentially be extended to multi-valued or continuous features with additional parametric and behavioral assumptions (e.g., partially linear models and agents misreporting $X^\ast$ by applying a fixed additive bias, such that $X = X^\ast + \delta$ for some constant $\delta$). We will add this as a promising direction for future work in the final version.
>
>
> **3 - What are the barriers to deploying CMRE in practice (e.g., data access and transparency)?** CMRE requires access to unmanipulated data -- typically available from pre-deployment training data or audits. Importantly, CMRE does not require access to the internals of the deployed model, making it applicable even with non-transparent, black-box decision models. We view the timeliness of our work as particularly conducive to its practical deployment, as it aligns with the need for operational efficiency and transparent models, increasingly expressed by AI regulators [3, 4].
>
>
> **4 - The paper could better explain why misreporting doesn't effect causal descendants before diving into technical details.** We thank the reviewer for this insightful suggestion. In the final version, we will add the following example based on our Medicare fraud use case: "If a patient truly has cancer ($X^\ast = 1$), they may experience cancer-related symptoms ($Y$). However, if an insurer misreports a cancer diagnosis ($X^\ast = 0$, $X = 1$), this misreporting does not cause the patient to exhibit cancer-related symptoms."
>
> **5 - [The work] identifies average misreporting rates but not individual ones.** We thank the reviewer for this comment, and wish to clarify that our method estimates a misreporting rate that is conditional on the agent identity rather than a population average. It can be trivially adapted to identify a misreporting rate that is conditional on features of individual datapoints (e.g., an individual's medical history in the Medicare setting), by estimating the conditional average treatment effect (CATE) versions of the TAFR, NAFR, and TAFM estimands in the paper, conditional on any features of interest. This allows us to identify which datapoints are most likely to be misreported.  We will clarify this in the final version.
>
>
> **6 - The reliance on causal descendants means the method may not apply to settings where downstream effects are unobservable or delayed.** The reviewer is correct that our method requires observing downstream outcomes. In practice, these are often available as the same labels used to train the manipulated model (e.g., enrollee costs in the Medicare risk-adjustment model example).
> When outcomes are delayed, surrogate or proxy outcomes can be used, consistent with existing causal literature [5]. We will clarify those two points in the final version.
>
>
> **7 - The approach is most useful in settings where agents have incentives to misreport and unmanipulated data is available, which may not cover all strategic ML scenarios.** We agree that our method is useful for settings where there is an incentive to misreport. Such scenarios are common in practice (e.g., healthcare insurance, hiring, and lending). We also note that unmanipulated data is often available from pre-deployment or from audits.
>
> **Citations**
>
> [1] Chang, Trenton, et al. "Who’s gaming the system? a causally-motivated approach for detecting strategic adaptation." Advances in Neural Information Processing Systems, 2024
>
> [2] Kallus, Nathan, Xiaojie Mao, and Angela Zhou. "Interval estimation of individual-level causal effects under unobserved confounding." The 22nd international conference on artificial intelligence and statistics. PMLR, 2019.
>
> [3] Winning the Race AMERICA’S AI ACTION PLAN, White House
>
> [4] Commission, European, et al. Adopt AI Study – Final Study Report. Publications Office of the European Union, 2024
>
> [5] Athey, Susan, et al. The surrogate index: Combining short-term proxies to estimate long-term treatment effects more rapidly and precisely. No. w26463. National Bureau of Economic Research, 2019.

---

### Note · Authors · 2025-08-11

We thank the reviewers for their constructive feedback and are encouraged by their recognition of our technical novelty and rigorous theoretical analysis. We have addressed all concerns as follows: Reviewers 9UrE and pJAD questioned the **no unobserved confounding assumption**. We clarified that it is standard in causality literature and that our approach is compatible with established sensitivity analyses that relax this assumption. Reviewer YpE8 raised concerns about **simulation robustness**. We added the reviewer’s suggested experiments, showing that our method remains robust across variations of our simulation. Reviewers YpE8 and 9UrE questioned our focus on **population-level misreporting rates**. We clarified that our method can be trivially extended to support more granular subgroup analysis by estimating the CATE version of our estimands conditioned on any features of interest. All reviewers asked about the **binary assumption for $X^{*}$ and $X$**. We explained that it simplifies defining misreporting rates and outlined that extensions to multi-valued or continuous features are possible under additional assumptions. These clarifications and results reinforce our claims, address the reviewers’ thoughtful questions, and make our paper even stronger.

---

### Decision · Program_Chairs · 2025-09-17

**Decision:**

Accept (poster)

**Comment:**

This paper introduces a causal inference framework to estimate the extent of misreporting in strategic settings, distinguishing deceptive reporting from genuine behavioral modifications. The approach leverages differences in causal effects on downstream variables across manipulated and unmanipulated datasets, with theoretical guarantees and empirical validation on synthetic and Medicare data.

The paper is well-regarded for its novelty in separating misreporting from genuine modification, the rigor of its identifiability analysis, and the policy relevance of its motivating applications. Reviewers noted that the framework is conceptually elegant, technically sound, and clearly presented, with the Medicare example demonstrating its practical importance.

Remaining concerns focused on modeling assumptions, estimator variance, and generalizability. Reviewers highlighted the reliance on the no unobserved confounding assumption, the restriction to binary features and one-directional misreporting, and the requirement for unmanipulated datasets. While these assumptions are strong, the rebuttal clarified their prevalence in causality work and offered sensitivity analysis and extensions as possible future work. Concerns about estimator variance were acknowledged, though reviewers agreed the method offers more reliable estimates than biased alternatives.

Given the novelty, theoretical contribution, clear positioning in the literature, and thoughtful empirical validation, the reviewers converged toward acceptance despite some minor remaining concerns such as variance of the estimator.

To further strengthen the work, the authors are encouraged to streamline the exposition of key results (e.g., moving Theorem 1 earlier), improve clarity around assumptions and their limitations, and expand on practical strategies for reducing estimator variance. Discussion of extensions to richer feature types and more complex misreporting patterns would also broaden the applicability of the framework.